# Genomic adaptations to aquatic and aerial life in mayflies and the origin of insect wings

Isabel Almudi [1✉], Joel Vizueta [2], Christopher D. R. Wyatt[3,4], Alex de Mendoza [5,6,7], Ferdinand Marlétaz [8], Panos N. Firbas[1], Roberto Feuda[9], Giulio Masiero[1], Patricia Medina [1], Ana Alcaina-Caro [1], Fernando Cruz [10], Jessica Gómez-Garrido [10], Marta Gut[10,11], Tyler S. Alioto [10,11], Carlos Vargas-Chavez [12], Kristofer Davie [13,14], Bernhard Misof[15], Josefa González [12], Stein Aerts [13,14], Ryan Lister [5,6], Jordi Paps [16], Julio Rozas [2], Alejandro Sánchez-Gracia [2], Manuel Irimia [3,11,17], Ignacio Maeso [1] & Fernando Casares [1✉]

The evolution of winged insects revolutionized terrestrial ecosystems and led to the largest animal radiation on Earth. However, we still have an incomplete picture of the genomic changes that underlay this diversification. Mayflies, as one of the sister groups of all other winged insects, are key to understanding this radiation. Here, we describe the genome of the mayfly *Cloeon dipterum* and its gene expression throughout its aquatic and aerial life cycle and specific organs. We discover an expansion of odorant-binding-protein genes, some expressed specifically in breathing gills of aquatic nymphs, suggesting a novel sensory role for this organ. In contrast, flying adults use an enlarged opsin set in a sexually dimorphic manner, with some expressed only in males. Finally, we identify a set of wing-associated genes deeply conserved in the pterygote insects and find transcriptomic similarities between gills and wings, suggesting a common genetic program. Globally, this comprehensive genomic and transcriptomic study uncovers the genetic basis of key evolutionary adaptations in mayflies and winged insects.

[1] GEM-DMC2 Unit, The CABD (CSIC-UPO-JA), Ctra. de Utrera km 1, 41013 Seville, Spain. [2] Departament de Genètica, Microbiologia i Estadística, Facultat de Biologia and Institut de Recerca de la Biodiversitat (IRBio), Universitat de Barcelona, Barcelona, Spain. [3] Centre for Genomic Regulation (CRG), The Barcelona Institute of Science and Technology, Barcelona, Spain. [4] Centre for Biodiversity and Environment Research, University College London, Gower Street, London WC1E 6BT, UK. [5] Australian Research Council Centre of Excellence in Plant Energy Biology, School of Molecular Sciences, The University of Western Australia, Perth, Western Australia, Australia. [6] Harry Perkins Institute of Medical Research, Perth, Western Australia, Australia. [7] Queen Mary University of London, School of Biological and Chemical Sciences, Mile End Road, E1 4NS London, UK. [8] Molecular Genetics Unit, Okinawa Institute of Science and Technology, Onna-son, Japan. [9] Department of Genetics and Genome Biology, University of Leicester, University Road, Leicester LE1 7RH, UK. [10] CNAG-CRG, Centre for Genomic Regulation (CRG), Barcelona Institute of Science and Technology (BIST), Baldiri i Reixac 4, 08028 Barcelona, Spain. [11] Universitat Pompeu Fabra (UPF), Barcelona, Spain. [12] Institute of Evolutionary Biology (IBE), CSIC-Universitat Pompeu Fabra, Barcelona, Spain. [13] Laboratory of Computational Biology, VIB Center for Brain and Disease Research, Herestraat 49, 3000 Louvain, Belgium. [14] Department of Human Genetics, KU Leuven, Oude Markt 13, 3000 Louvain, Belgium. [15] Zoological Research Museum Alexander Koenig, Adenauerallee 160, 53113 Bonn, Germany. [16] School of Biological Sciences, University of Bristol, 24 Tyndall Avenue, Bristol BS8 1TQ, UK. [17] ICREA, Barcelona, Spain. ✉email: isabelalmudi@gmail.com; fcasfer@upo.es

The first insects colonized the land more than 400 million years ago (MYA)[1]. But it was only after insects evolved wings that this lineage (Pterygota) became the most prominent animal group in terms of number and diversity of species, and completely revolutionized Earth ecosystems. The development of wings also marked the appearance of a hemimetabolous life cycle[2], with two clearly differentiated living phases (non-flying juveniles and flying adults). This allowed pterygote insects to specialize functionally and exploit two entirely different ecological niches. This is still the life cycle of Ephemeroptera (mayflies) and Odonata (damselflies and dragonflies), the extant Paleoptera ("old wing") orders, the sister group of all other winged insects (Fig. 1a). All paleopteran insects undergo a radical ecological switch, where aquatic nymphs metamorphose into terrestrial flying adults. The appearance of wings and the capacity to fly greatly increased the capability of insects for dispersal, escape and courtship and allowed them access to previously unobtainable nutrient sources, while establishing new ecological interactions. This 'new aerial dimension in which to experience life'[3] created new evolutionary forces and constraints that since, have been continuously reshaping the physiology, metabolism, morphology and sensory capabilities of different pterygote lineages—evolutionary changes that should be mirrored by modifications in their genomes. However, our knowledge of these genomic changes is still incomplete, mostly because paleopteran lineages (Fig. 1a), have not been extensively studied from a genomic and transcriptomic perspective.

Mayflies are an ideal group to fill this gap. By living in both aquatic and terrestrial environments, mayflies had to develop different sensory, morphological and physiological adaptations for each of these ecological niches. For example, mayflies have abdominal gills during the aquatic stages, a feature that places them in a privileged position to assess the different hypotheses accounting for the origin of wings, which suggested that wings are either homologous to tergal structures (dorsal body wall), or pleural structures (including gills) or a fusion of the two[4–9]. Moreover, some mayfly families exhibit a striking sexual dimorphism in their visual systems, which in the case of the Baetidae family, includes the presence of a second set of large compound eyes in males (Fig. 1d, e). All these features make mayflies an excellent order to investigate the origin of evolutionary novelties associated with the conquest of new habitats. The recent establishment of a continuous culture system of the mayfly *Cloeon dipterum*, a cosmopolitan Baetidae species with a life cycle of just 6 weeks, makes it now possible to access all embryonic and post-embryonic stages. This overcomes past difficulties to study paleopterans which are generally not very amenable to rear in the laboratory; with mayflies having the additional challenge of being extremely short-lived as adults[10].

Here we sequence the genome and profile a comprehensive set of stage- and organ-specific transcriptomes of *C. dipterum*. Our analyses identify potential genomic signatures associated with mayfly adaptations to an aquatic lifestyle, innovations in its visual system and novel genetic players in the evolution of wings. The results from this work establish *C. dipterum* as a new platform to investigate insect genomics, development and evolution from a phylogenetic vantage point.

## Results

**C. dipterum genome and transcriptome assemblies**. We sequenced and assembled the genome of an inbred line of the mayfly species *C. dipterum* using both Illumina and Nanopore technologies (see Methods, Supplementary Fig. 1, Supplementary Data 2–4). The *C. dipterum* genome was assembled in 1395 scaffolds, with an N50 of 0.461 Mb. The total genome assembly length of *C. dipterum* is 180 Mb, which in comparison to other pterygote species[11–14], constitutes a relatively compact genome, probably due to the low fraction of transposable elements (TEs) (5%, in contrast to the median of 24% ± 12% found in other insects[15], Supplementary Fig. 1). The gene completeness of the genome was estimated to be 96.77 and 98.2% (94.1% complete single-copy, 2.8% complete duplicated and 1.3% fragmented BUSCOs), according to Core Eukaryotic Genes Mapping Approach (CEGMA v2.5[16]) and Benchmarking Universal Single-Copy Orthologs (BUSCO v3[17]), respectively. Moreover, the fact

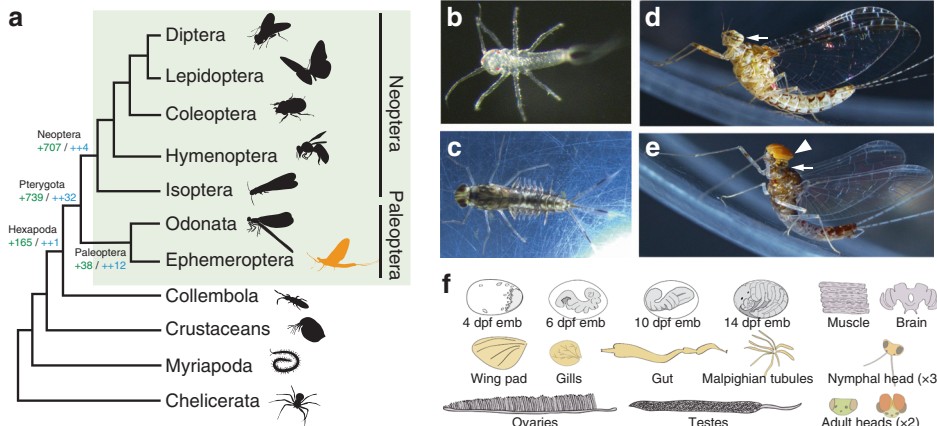

**Fig. 1 The mayfly C. dipterum. a** Simplified arthropod phylogeny highlighting Pterygota insects (green rectangle). Ephemeroptera (mayflies, in orange) are within Paleoptera, the sister group to all other winged insects. Number of gene gains (in green) and core gene gains (in blue) is shown at the root of selected lineages. **b** *C. dipterum* freshly hatchling. **c** Late nymph. **d** Female adult. **e** Male adult. **f** Cartoons depicting organs and developmental stages used for the transcriptome profiling. In grey, embryonic stages used: 4 days post fertilization (dpf) stage: germ disc (2 replicates, Paired-End (PE) RNA samples); 6 dpf stage: segmentation (1 replicate, PE); 10 dpf stage: revolution (2 replicates, PE) and 14 dpf stage: pre-nymph (2 replicates, PE)); in orange, nymphal tissues: heads of three different nymphal stages (4 replicates young/early nymphs, 2 replicates mid-stage nymphs, 4 replicate late nymphs (2 female and 2 male), Single-End (SE) RNA samples), nymphal gut (×2, PE), nymphal Malpighian tubules (×2, PE), gills (×2, PE), wing pads (×2, PE); pale pink: adult muscle (×2, PE), ovaries (×2, PE), testes (×2, PE), female adult brain (×2, PE) and male and female adult heads (2× each, PE, see Supplementary Data 7 for sample details). White arrows in (**d**) and (**e**) highlight compound eyes present in both male and female, whereas the turbanate eye (white arrowhead) is only present in the male (**e**).

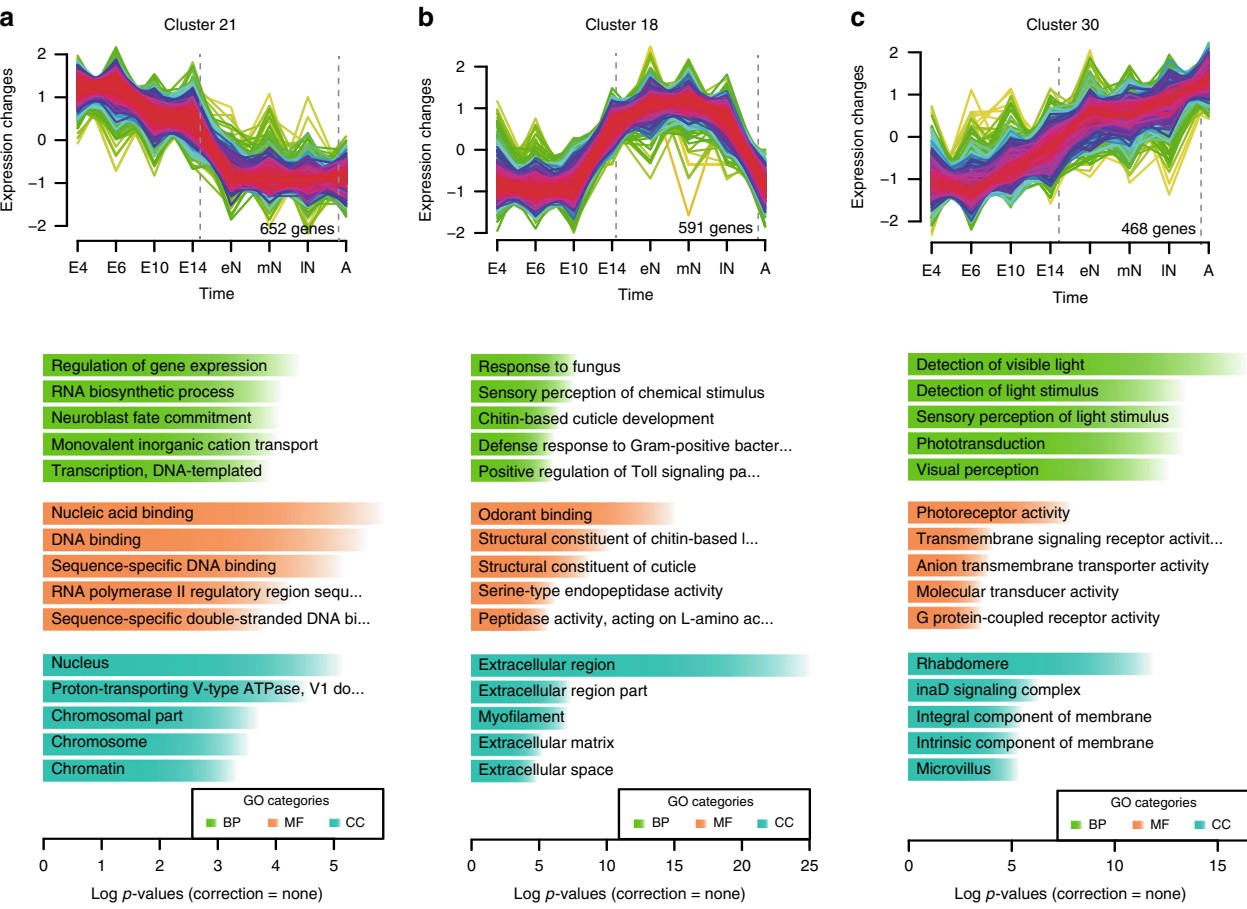

**Fig. 2 C. dipterum transcriptomes throughout its life cycle. a** Mfuzz clusters showing high expression during embryonic, **b** nymphal and **c** adult stages (E4: 4 dpf embryo, E6: 6 dpf embryo, E10: 10 dpf embryo, E14: 14 dpf embryo, eN: early nymph head, mN: mid nymph head, lN: late nymph head, A: adult head). Plot numbers indicate the number of genes assigned to each of these clusters. Colour code reflects 'Membership' values calculated by Mfuzz, where magenta corresponds to high values and green to low values of Membership score. Histograms show enriched GO terms (determined with topGO R package). Each of them contains genes that are enriched in GO terms related to developmental processes, chitin and chemical stimuli perception and visual perception, respectively. Uncorrected *P* values shown in plots corresponded to two-sided Fisher's exact tests (BP: biological process, CC: cellular component, MF: molecular function).

that we have found 94.1% complete single-copy and only 2.8% complete duplicated BUSCOs, out of 1658 insect orthologues (see Supplementary Information), and the seven generations of inbreeding of the individuals used for the DNA extraction, supports the haploid nature of our reference genome. Protein-coding gene annotation resulted in 16,730 genes, which were used to reconstruct orthologous gene families of *C. dipterum* with other animal species (Supplementary Data 5 and 6).

Along with the genomic DNA, we transcriptionally profiled a large number of time points along *C. dipterum's* life cycle and a set of nymphal and adult organs. These datasets included four embryonic stages (4 days post fertilization (dpf) stage: germ disc); 6 dpf stage: segmentation; 10 dpf stage: revolution and 14 dpf stage: pre-nymph, heads of three different nymphal stages, nymphal gut, nymphal Malpighian tubules, gills, wing pads, adult muscle, ovaries, testes, female adult brain and male and female adult heads (Fig. 1f, see Fig. 1 legend, Supplementary Data 7 and Supplementary Note 1 for details on the number of replicates and sequencing).

**Gene expression dynamics reflect life cycle adaptations.** *C. dipterum* spends its life cycle in three different environments: within the abdomen of the mother during embryonic stages (as *C. dipterum* is one of the few ovoviviparous mayfly species[16]);

freshwater streams and ponds as nymphs; and land/air as adults[17]. To explore gene expression patterns during these three major phases, we performed a temporal soft-clustering analysis of stage-specific transcriptomes using Mfuzz[18] and focused on clusters containing genes whose expression peaks at each of these phases (Fig. 2, Supplementary Fig. 2, Supplementary Data 8 and 9). Clusters of genes transcribed preferentially during embryonic stages, such as cluster 21 and 8, showed enrichment in Gene Ontology (GO) terms that reflected the processes of embryogenesis and organogenesis happening during these stages (i.e. regulation of gene expression, neuroblast fate commitment, DNA binding, etc., Fig. 2a). On the other hand, clusters with genes highly expressed during the aquatic phase (e.g., cluster 18, Fig. 2b) presented GO enrichment in terms consistent with the continued moulting process that mayfly nymphs undergo, such as chitin-based cuticle development and defence response[19,20]. In addition, GO terms related to sensory perception of chemical stimuli or odorant binding were also enriched in these clusters (Fig. 2b, Supplementary Fig. 2, Supplementary Data 9). Finally, cluster 30, which contained genes with the highest expression during adulthood showed a striking enrichment of GO terms associated with visual perception (Fig. 2c). Therefore, the embryonic and post-embryonic (aquatic and aerial) phases are characterized by distinct gene expression profiles. Specifically,

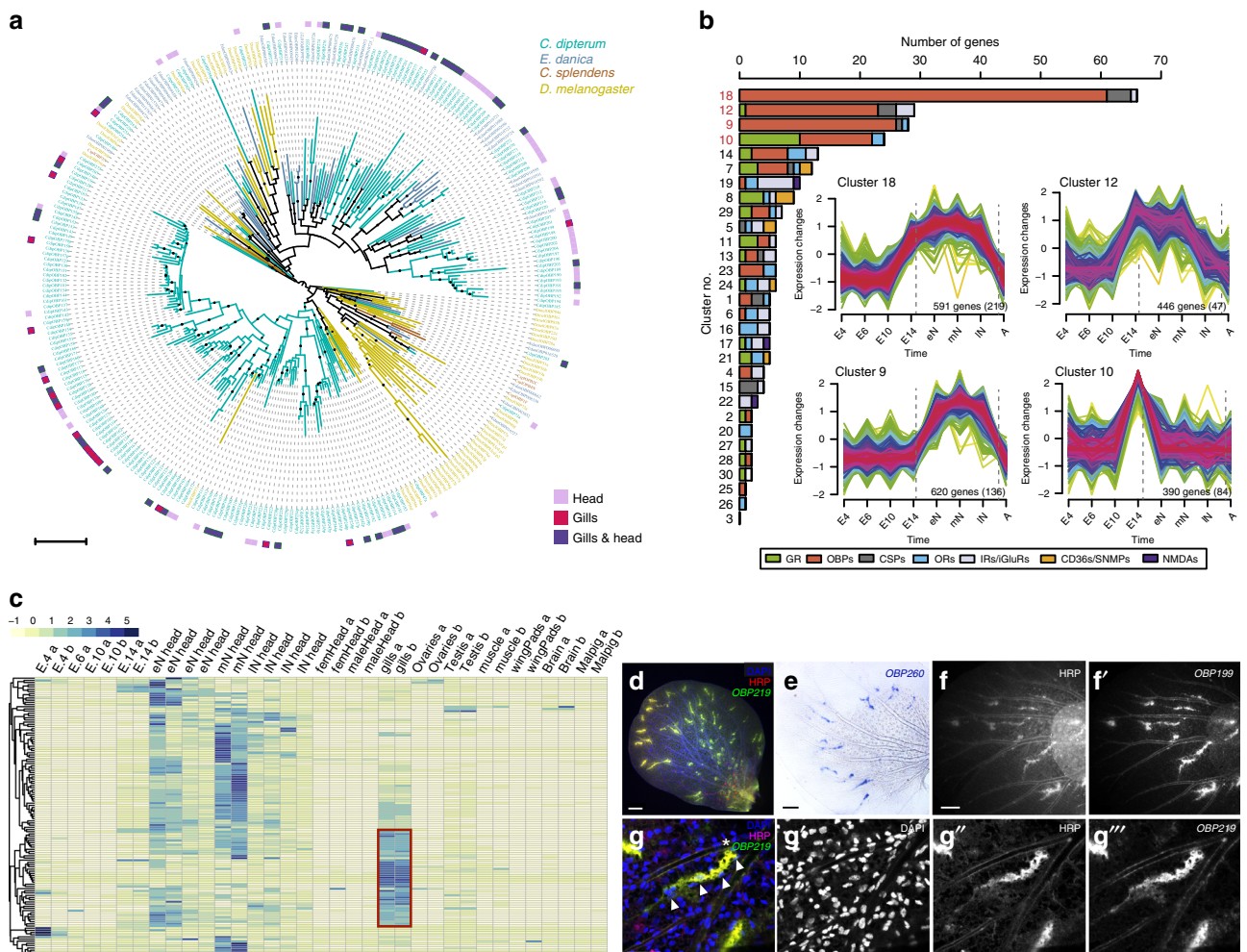

**Fig. 3 OBP gene family in *C. dipterum*. a** OBP phylogenetic reconstruction. *C. dipterum* genes are shown in green, *E. danica* in blue, *C. splendens* in brown and *D. melanogaster* in mustard. OBPs expressed specifically (according to tau values > 0.8; Supplementary Data 11) in nymphal heads are highlighted by pink squares, in gills in magenta and OBPs expressed both in heads and gills are shown in purple. **b** *C. dipterum* chemosensory families distributed across Mfuzz clusters. Clusters containing the largest amounts of chemoreceptors are no. 18, 12, 9 and 10 (E4: 4 dpf embryo, E6: 6 dpf embryo, E10: 10 dpf embryo, E14: 14 dpf embryo, eN: early nymph head, mN: mid nymph head, lN: late nymph head, A: adult head). Numbers in plots show the genes, and between parentheses, core genes for each cluster. **c** Heat map showing high levels of OBP expression across RNA-seq samples. Increased expression in nymphal head samples. An important fraction of OBP genes are highly expressed in gills (red square). **d** Spatial expression of *OBP219* (green) in neural structures, marked by HRP staining (in red) in abdominal gills. **e** Spatial expression of *OBP260* in abdominal gills. **f–f′**, *OBP199* expression pattern in the gills (**f′**) co-localises with HRP-stained neural cells. **g** Detail of a cluster of cells expressing *OBP219* (**g‴**) and marked by HRP (**g″**) in close contact with a trachea (highlighted by white asterisk). Nuclei are detected by 4′,6-diamidino-2-phenylindole (DAPI) staining (white arrowheads). Scale bars: 50 μm. Spatial expression of OBP genes (**d–g‴**) was assayed in at least two independent experiments with n > 20 gills each.

comparison between aquatic nymphs and aerial adults highlights expression differences that indicate distinct sensory modalities in these two free-living phases. The aquatic phase is characterised by genes involved in the perception of chemical cues, whereas vision becomes the main sensory system during the terrestrial/aerial adult phase (Fig. 2).

**Role of odorant-binding protein genes during aquatic phase.** Since temporal gene expression profiling indicated a prominent role of genes involved in perception of chemical cues during *C. dipterum* aquatic phase, we investigated in its genome the five main chemosensory-related (CS) gene families in arthropods. These include the odorant receptor/odorant receptor coreceptor (OR/ORCO) and the odorant-binding protein family (OBP), both of which have been suggested to play essential roles during the evolution of terrestrialization in hexapods and insects[21–27].

We identified 64 gustatory receptor (GR), 23 Ionotropic receptors (IR), 3 N-methyl-d-aspartate (NMDA) genes, 12 CD36/SNMP, 16 chemosensory proteins (CSP), one copy of ORCO and 49 OR genes in the *C. dipterum* genome. The size of CS gene families is similar to the size of the damselfly *Calopteryx splendens*[11], with the exception of *C. splendens* OR and OBP gene families, which are much smaller (Supplementary Fig. 3, Supplementary Data 10). Indeed, in the case of the OBP family, we discovered 191 different OBP genes (of which 167 have complete gene models), which represents the largest repertoire of this family described until now, and a 75% increase with respect to the largest OBP gene complement previously described, that of the cockroach *Blattella germanica* (109 OBP genes)[22,28] (Fig. 3a, Supplementary Fig. 3).

Our GO term enrichment analyses of temporal co-expression clusters pointed to an important role of these CS genes during nymphal stages. To investigate this further, we asked in which

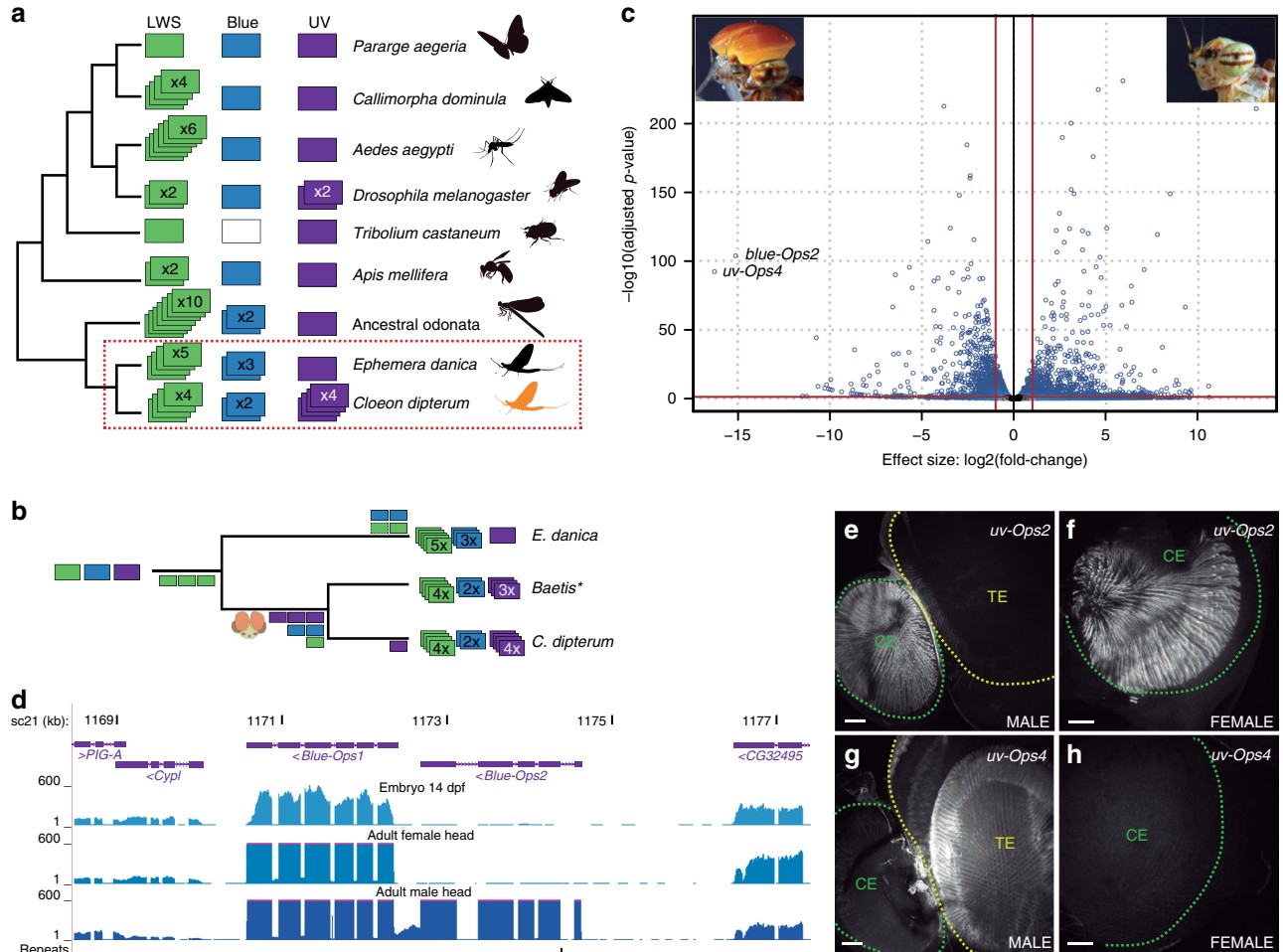

**Fig. 4 Opsin gene family in *C. dipterum*. a** opsin gene complement in insect phylogeny ('ancestral odonata' refers to the Opsin complement of the last common ancestor of this lineage as inferred in ref. [72]). **b** Evolutionary scenario for the expansions of three families of opsins in mayflies. **c** Volcano plot showing differential gene expression between male and female adult heads. **d** Blue-Opsin cluster in the *C. dipterum* genome browser. Upper and middle track show expression of *Blue-Ops1* in late embryos (st. E14 dpf) and adult female heads, while lower track shows high expression of both *Blue-Ops1* and *Blue-Ops2* in male adult heads. **e, f** Expression of *UV-Ops2* in the compound eye of a male (**e**) and a female (**f**). **g, h** *UV-Ops4* high expression in the turbanate eye of a male (**g**) is not detectable in the female (**h**). TE: turbanate eye, CE: compound eye. Scale bars: 50 μm. Ops gene expression was detected in three independent experiments with *n* > 5 retinas each.

clusters the different CS gene families were incorporated. We found that more than half of CS genes, and among these, nearly 80% of OBPs (147 out of 276 CS genes and 121 out of 152 OBPs included in the soft-clustering analysis) were mostly assigned to just four clusters (18, 12, 9 and 10), all of them related to nymphal or pre-nymphal stages. Cluster 10 contained genes that had a peak of expression at 14 dpf stage, just prior to the hatching of the first swimming nymph[10], and genes in clusters 9, 12 and 18 were most highly expressed during nymphal stages (Fig. 3b).

Next, we examined the expression of CS genes in specific tissues and organs and found that most of the CS genes were expressed in a highly tissue-specific manner, as indicated by high (>0.8) tau values (an index used as a proxy for tissue-specific expression[29], Supplementary Data 11). As expected, many CS genes were expressed in the head, where the antennae, the main olfactory organs in insects, are located. There was however, an additional major chemosensory tissue, the gills, where 34% of the 276 CS genes were expressed, (5/16 CSP, 8/34 IR and 82/167 OBP genes, Fig. 3c, Supplementary Fig. 4). These included 25 gill specific OBPs, several of which (*OBP219*, *OBP199* and *OBP260*) were shown to be expressed in discrete cell clusters within the

gills by in situ hybridization, often located at the branching points of their tracheal arborization (Fig. 3d–g). To better characterise these clusters, we co-stained the gills with horseradish peroxidase (HRP), a marker of insect neurons[30]. Remarkably, OBP-expressing cell clusters were HRP-positive, suggesting a neuro-sensory nature. Globally, these results strongly suggest that, beyond their respiratory role[31], the gills are a major chemosensory organ of the aquatic mayfly nymph (Fig. 3f, g).

**Expansion of light-sensing opsin genes in *C. dipterum*.** Gene expression dynamics and GO enrichment analyses showed that visual perception must play a prominent role during adulthood in mayflies (Fig. 2c). During their short adult phase, they must be able to find mating partners while flying in large swarms, copulate and finally females ought to find a suitable place to deposit the eggs[32].

Similar to what has been observed in other diurnal insect lineages, such as Odonata[33], we found an expansion of long wave sensitive (LWS) opsin genes in the *C. dipterum* genome, with four LWS opsin duplicates grouped in a genomic cluster (Fig. 4a, b,

Supplementary Fig. 5). This LWS cluster was also present within Ephemeroptera in the distantly related Ephemeridae, *Ephemera danica*, emphasizing the importance that these light-sensing molecules have had in the evolution and ecology of the entire mayfly group (Supplementary Fig. 5). In addition, we also found that the blue-sensitive opsin underwent independent duplications in the Ephemeridae and Baetidae. *E. danica* has three different *blue-Ops*, while *Baetis* species with available transcriptomic data (*B. sp.* AD2013, *B. sp.* EP001, ref. [1]), and *C. dipterum* have two *blue-Ops*, which in the latter case are located together in tandem (Fig. 4a, d).

Surprisingly, and in contrast to other insect lineages, where the ultraviolet (UV) sensing opsin (*UV-Ops*) has usually been kept as a single copy, we also found four copies of UV-Ops in the genome of *C. dipterum*. Duplicated UV-Ops genes were also present in different *Baetis* species (Fig. 4a, b and Supplementary Fig. 5), indicating that the baetid last common ancestor had at least three copies of UV-Ops genes, while *C. dipterum* acquired an extra duplicated *UV-Ops4*. This makes this species' UV-Ops complement the largest one described thus far (Fig. 4a).

The most salient feature of the Baetidae family is the presence of a sexually dimorphic pair of large extra compound eyes on the dorsal part of the head, called the turbanate eyes, which develop during nymphal stages only in males[34] (Fig. 1e). Indeed, the most highly upregulated gene in adult male heads compared to female heads was one of the UV-sensitive opsins, *UV-Ops4* followed by *blue-Ops2* (Fig. 4c, d, Supplementary Data 12), a divergent blue-Ops duplicate comprising only the N-terminus part of the protein containing the retinal binding domain. Moreover, *UV-Ops4* and *blue-Ops2* expression was only detectable from late nymph stages onwards, whereas most other opsins were expressed already in the pre-nymphal stage (14 dpf embryo, when the lateral compound eyes and ocelli, common to both males and females, start developing; Fig. 4d). Their delayed expression onset and their sexual dimorphism strongly suggested that both *UV-Ops4* and *blue-Ops2* are turbanate eye-specific opsins, while the rest of *C. dipterum* opsins function in the lateral compound eyes and/or ocelli of males and females. Consistently, in situ hybridization assays showed that the shared *UV-Ops2* was expressed in the compound eye of both sexes (Fig. 4e, f), while *UV-Ops4* was predominantly expressed in the turbanate eyes of males but undetectable in females (Fig. 4g, h). Thus, this sexually dimorphic opsin system may be of particular relevance for courtship and mating during flight.

**A conserved core set of wing genes in pterygote insects**. The terrestrial/aerial adult phase of *C. dipterum* is not only characterised by its visual system, but even more prominently by the key feature ancestral to all pterygote insects: the wings.

As paleopterans, mayflies can provide important insights into the origin of the genetic programmes responsible for the evolution of this morphological novelty. To investigate this, we first generated modules of co-regulated gene expression across several tissues, using Weighted Gene Correlation Network Analysis (WGCNA, Supplementary Data 13, Supplementary Data 1), including developing wings, for *C. dipterum* and *D. melanogaster* (Fig. 5b and see Supplementary Note 3, Supplementary Data 1). To address transcriptomic conservation, we tested which modules showed significant orthologue overlap in pairwise comparisons between the two species. This analysis revealed deeply conserved co-regulated gene modules associated with muscle, gut, ovaries, brain and Malpighian tubules (insect osmoregulatory organ), among others, indicating that the shared morphological features of the pterygote body plan are mirrored by deep transcriptomic conservation (Fig. 5a). Remarkably, when

we extended this comparison to include the centipede *Strigamia maritima*, the majority of these co-regulated modules, especially those corresponding to neural functions, muscle and gut, have been conserved as well, indicating that these genetic programmes date back, at least, to the origin of Mandibulata (Supplementary Fig. 6a). Notably, we also found this deep conservation using an alternative metric for module preservation and in-group proportion statistics[35–38], adding further support to our results (Supplementary Fig. 6b).

Importantly, one of the highly correlated pairs of modules between mayfly and fruitfly corresponded to the wings (wing pad and wing imaginal disc modules, respectively). A total of 126 orthologous gene families were shared between these modules, defining a core set of genes that could have been associated with this organ since the last common ancestor of pterygote insects (corresponding to 130 and 128 genes in *D. melanogaster* and *C. dipterum*, respectively). This gene set exhibited an enrichment in GO terms such as morphogenesis of a polarized epithelium, consistent with wing development (Fig. 5c), and in agreement with this, some of these orthologues have been shown to participate in wing development in *Drosophila* (e.g. *abnormal wing discs* (*awd*), *inturned* (*in*) or *wing blister* (*wb*), etc.). In addition, among them there were also numerous genes (96 genes out of 130, Supplementary Data 14) for which no previous wing function has been described so far. However, our comparative approach strongly indicated a putative function in wing development. We functionally tested this hypothesis for eight of these genes in *Drosophila* (Supplementary Data 15) by RNAi knockdown assays using the wing-specific *nubbin-GAL4* driver[39]. Indeed, these experiments resulted in wing phenotypes in all cases (8/8) and in particular, abnormal wing venation (7/8) (Fig. 5d–f, Supplementary Fig. 6c).

We next asked if the origin of novel traits in particular lineages, including the wings in pterygotes, might have been coupled to the appearance of new genes. To this end, we classified orthologous genes according to their evolutionary origin and checked whether there was any correlation between their appearance and the tissues in which they were mainly expressed, based on the results obtained through the WGCNA modules (Fig. 5a, Supplementary Data 1) and tau index analyses (Supplementary Fig. 6d and Supplementary Data 11). In general, our results were in agreement with previously described phylostratigraphic patterns, in which older genes exhibit broad expression patterns while novel genes are expressed in a tissue-specific manner[40,41]. Moreover, we studied gene gains and losses at the origin of winged insects (see Methods, Fig. 1a and Supplementary Fig. 1). We did not observe any enrichment of new genes expressed in the wings in the Pterygota stratum that could be linked to the origin of this morphological novelty, despite the high number of novel genes that were gained in this lineage (Fig. 1a, Supplementary Fig. 1). These results suggested that the transcriptomic programme responsible for the wings was assembled from genes that were already forming part of pre-existing gene networks in other tissues[42]. Therefore, we investigated the similarities between the transcriptomic programmes of wings and other mayfly tissues (Fig. 5g, h). Clustering *C. dipterum* genes based on their expression across the different tissues, organs and developmental stages revealed that gills were the most closely related organ to (developing) wings (Fig. 5g). In fact, gills were the tissue that shared most specific genes with wing pads (42/98, 43%), as assessed by looking for genes with high expression only in wings and an additional tissue (see Methods, Fig. 5h, Supplementary Fig. 6e, f, Supplementary Data 16), supporting the transcriptional similarities between gills and wing pads in mayflies. Moreover, when checking the function of these *Drosophila* orthologs, 40% (12/30) of them had a known role in wing development (e.g., *shifted* and *taxi*).

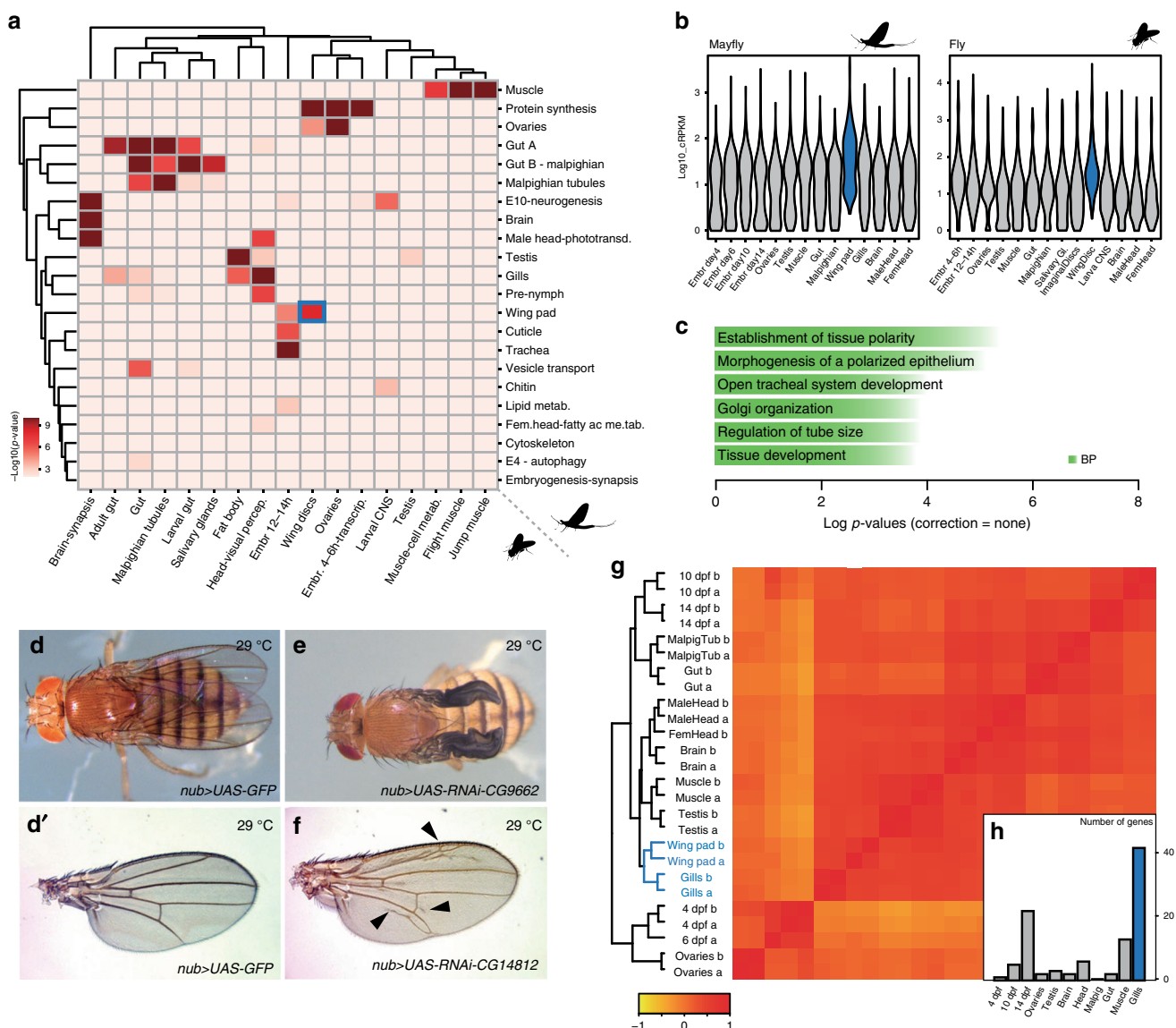

**Fig. 5 Transcriptomic conservation of wings and other insect tissues. a** Heatmap showing the level of raw statistical significance, using upper-tail hypergeometric tests, of orthologous gene overlap between modules from *C. dipterum* (vertical) and *Drosophila* (horizontal) obtained by Weighted Gene Correlation Network Analysis (WGCNA). **b** Gene expression distributions (log10 cRPKMs) within the wing modules across each sample (**a**) for *C. dipterum* (left) and *D. melanogaster* (right). **c** Enriched Gene Ontology terms (BP: biological process) of the orthologous genes shared between *C. dipterum* and *D. melanogaster* wing and wing disc modules, using the wing disc module of *D. melanogaster* as background. **d–f** Wing phenotypes resulted from knockdowns of orthologous genes shared between *C. dipterum* and *D. melanogaster* wing and wing disc modules using wing-specific driver *nub-Gal4*. **d**, **d′** *nub-Gal4; UAS-GFP* control fly (**d**) and control wing (**d′**). **e** *nub-Gal4; UAS-RNAi-CG9662* flies show severe wing phenotypes. **f** *nub-Gal4; UAS-RNAi-CG14812* show extra vein tissue phenotypes. **g** Heat map showing transcriptome datasets clustering. *C. dipterum* wing pads and gills samples cluster together. **h** Number of genes per each of second most tissues for 98 genes highly expressed in wing pads (*n* = 98). Gills are highlighted in blue.

## Discussion

Together with the possibility of culturing *C. dipterum* continuously and its life cycle of about 45 days, the *C. dipterum* genome sequence and transcriptome datasets presented here provide the foundations for the exploration of a number of important evolutionary, developmental and physiological aspects of the biology of insects. When interrogated with respect to gene expression associated with the two disparate environments that mayflies adapted to (aquatic and aerial), the data suggest a sensory specialization, with nymphs predominantly using chemical stimuli, whereas adults rely predominantly on their visual system.

Arthropods perceive different chemical environmental cues, such as pheromones, food or the presence of predators, using different families of chemosensory proteins specially tuned to the

structural and chemical characteristics of those cues (e.g. volatile, hydro soluble, etc.). Accordingly, CS families in arthropods have undergone multiple lineage-specific changes based on adaptations to the myriad of ecological niches they occupy. The possible co-option of OBPs by Hexapoda[24] and the appearance of ORs in insects led to the idea that these two families evolved as an adaptation to terrestrial life and that their main function is to perceive smells in the aerial media. However, the role of many OBP genes is still unclear, and functions other than olfaction have been proposed for OBPs in terrestrial insects[43]. The finding of OBP-expressing organs exhibiting neural markers in gills—apart from the heads—and associated with the trachea, together with the specific expression of some chemoreceptors in this tissue (Supplementary Fig. 4) indicates that gills may be a prominent

chemosensory organ in the mayfly. In fact, their large combined surface makes them especially apt for this function. Further, the presence of these chemosensory structures in the gills challenges the classic idea of gills as exclusively respiratory organs[31].

Living in two media (water and air) and the ability to fly must have imposed a number of specific requirements on the mayfly visual system, from vision in two refractive index media, or the use of novel visual cues for mating in the air, to an increase of the speed of visual information flow. For instance, insects evolved their visual systems and different types of light-sensing opsins to navigate during sunlight or moonlight illumination or to use polarized light to obtain important information from the environment[33,44]. Multiple insect lineages expanded the LWS opsin complement. In a similar way, Ephemeroptera show an ancestral expansion of LWS opsins, located in a cluster. In contrast, most insect groups have kept blue and UV opsins as single-copy genes (Fig. 4a). However, we found that the Baetidae family, which includes *C. dipterum*, has duplicated these two light-sensing gene families, with the largest UV opsin expansion described so far. The duplication of this particular type of opsins in Baetidae could be related to the origin and evolution of the male-specific turbanate eyes in this family of mayflies, since we observed that *UV-Ops4* is exclusively expressed in this novel sexually dimorphic visual system (Fig. 4g). UV opsins are usually more prominent in the dorsal part of insect eyes, due to their direct and frontal/upward position towards the sunlight[44–46]. Since it has been suggested that males use their dorsal turbanate eyes to locate females flying above their swarm, sexual selection might have played an important role during the evolution of UV and blue-opsin duplications and their specific expression in turbanate eyes of males.

Our transcriptomic comparisons between *C. dipterum*, *Drosophila* and *Strigamia* revealed for the first time high conservation of gene expression in multiple homologous arthropod tissues and organs, indicating that deep conservation of transcriptomic programmes may be a common signature in different phyla[47,48]. We identified a core set of genes associated to wing development that are probably ancestral to all pterygote insects. Future studies in additional pterygote lineages and developmental stages will help to refine this set and to identify potential cases of genes that could have been independently recruited to wing development in mayflies and *Drosophila*. Nevertheless, these 126 genes constitute the first genome-wide glimpse into the ancestral genetic make-up of this key morphological novelty. Notably, when we tested functionally some of these genes with no previously described functions, all of them showed wing phenotypes, and most often defects in veins. Veins are thickenings of the wing epithelium that provide it with sufficient rigidity for flight and which are unique anatomical wing features, consistent with the specificity of these transcriptional programmes and the late nymphal stages of our wing pad RNA samples. Overall, these results attest to the strong predictive power of our evolutionary comparative approach to infer function of conserved genes.

Several hypotheses to explain the origin of wings have been put forward since the last century[4,5,7–9,49], including the pleural origin hypothesis in which the wings would be the thoracic serial homologues of the abdominal gills. Here, we have uncovered transcriptomic similarities between gills and wings. As the first genome-wide expression comparison of these mayfly organs, further work including additional appendages and developmental stages as well as additional mayfly species will be needed to better understand the biological meaning of these similarities. Regardless, our results would be consistent with both pleural and dual scenarios, in which ancestral thoracic gills would contribute to wing structures. Alternatively, in a scenario in which gills and the aquatic nymphs would not be the ancestral state to pterygotes[50,51]

these similarities could represent gene co-option between these two organs. Whatever the case, the transcriptomic similarities observed between gills and wings suggest that they share a common genetic programme.

## Methods

**Genome sequencing and assembly**. Genomic DNA was extracted from *C. dipterum* adult males from an inbred line kept in the laboratory for seven generations[10]. Illumina and Nanopore technologies were used to sequence the genome (Supplementary Data 2–4 and Supplementary Note 1). The assembly was generated using the hybrid approach of Maryland Super-Read Celera Assembler (MaSuRCA) v3.2.3[52,53] with 95.9x short-read Illumina and 36.3x Oxford Nanopore Technologies (ONT) long-read coverage.

**RNA sequencing and assembly**. 37 RNA-seq datasets (including replicates) of multiple developmental stages (four embryonic stages) and dissected tissues and organs (nymphal and adult dissected organs and whole heads) were generated using the Illumina technology. Samples were processed immediately after dissection and RNA was extracted using RNeasy Mini Kit (Qiagen) or RNAqueous™-Micro Total RNA Isolation Kit (Ambion) following manufacturers' instructions. Single-end and paired-end libraries were generated using Illumina (TruSeq) RNA-Seq kit (see Supplementary Note 1 for more details).

**Genome annotation**. Reads from all paired-end transcriptomes were assembled altogether using Trinity[54] and subsequently aligned to the genome using the Programme to Assemble Spliced Alignments (PASA) pipeline[55]. High-quality transcripts were selected to build a Hidden-Markov profile for de novo gene prediction in Augustus[56].

Reads were aligned to the genome using Spliced Transcripts Alignment to a Reference (STAR)[57] and transcriptome assembled using Stringtie[58]. The assemblies were merged using Taco[59]. Consensus transcriptome assembly and splice-junctions were converted into hints and provided to Augustus gene prediction tool which yielded 16364 evidence-based gene models. These models contain 4308 non-redundant PFAM models as assigned using the PfamScan tool. RepeatModeler[60] was used to identify repeats in *C. dipterum* assembled genome (Supplementary Note 1).

We integrated these RNA-seq data into a University of California Santa Cruz (UCSC) Genome Browser track hub together with two additional tracks, insect sequence conservation and annotation of repetitive elements.

**Gene orthology**. To obtain phylogeny-based orthology relationships between different taxa, the predicted proteomes of 14 species (Supplementary Data 5) representing major arthropod lineages and outgroups were used as input for OrthoFinder2[61] (Supplementary Note 2).

**Comparative transcriptomics**. Mfuzz software[18] was used to perform soft clustering of genes according to developmental and life history expression dynamics using normalised read counts. We selected eight developmental and post-embryonic stages.

We used DESeq2 R package[62] to analyse differential gene expression between male and female adult heads.

We used the cRPKM metric (corrected-for-mappability Reads Per Kilobasepair) of uniquely mappable positions per Million mapped reads[63] to perform WGCNA gene expression analyses. We performed the analyses using as datasets genes that were present in a least two species in our family reconstructions and showed variance across samples (coef. var ≥ 1). Each module was designated with a tissue or a biological/molecular category. Finally, we analysed the overlap between homologous groups for each pair of modules for each of the species in a pairwise manner. To evaluate significance, we performed hypergeometric tests (see details in Supplementary Note 3).

**GO term assignment and enrichment analyses**. Taking the orthology groups from *C. dipterum* or *Strigamia maritima* to *D. melanogaster* GO database from Ensembl BIOMART, we generated a topGO gene to GO key, by copying across all GO terms represented in each orthogroup. We performed an Enrichment Analysis for Gene Ontology using topGO package (v 2.36.0)[64] Uncorrected $P$ values shown in plots corresponded to two-sided Fisher's exact tests (see Supplementary Note 1 for details).

**Chemosensory gene identification and phylogeny**. We created a dataset containing well-annotated members of the chemosensory (CS) gene family (i.e., GR, OR, IR/iGluR, OBP and CSP) from a group of representative insects (Supplementary Data 10[11,13,26–28,65–69]). In addition, we constructed specific Hidden-Markov Models (HMM) profiles for each CS gene family based on their Pfam profiles (see Supplementary Table 1 in ref. [70]). Briefly, we performed iterative rounds of BLASTP and HMMER searches against the annotated proteins of *C. dipterum*, curating incorrect gene models, and TBLASTN against the genomic

sequence to identify CS proteins. Finally, we obtained a curated GTF containing the annotation for each CS gene family (see Supplementary Note 4 for details).

**In situ hybridizations**. Specific primers were designed to generate DIG-labelled RNA probes against *OBP260, OBP219, OBP199, UV-Ops2* and *UV-Ops4* (Supplementary Data 17). After overnight (o.n.) fixation of the gills or heads in 4% FA at 4 °C, post-fixated gills and retinas were treated with Proteinase K for 15 min. The hybridization was carried out at 60 °C o.n. Tissues were incubated with anti-digoxigenin-POD at 4 °C o.n. and with 1:100 TSA in borate buffer for 1 h. Leica SPE confocal microscope was used to acquire images that were processed with Fiji[71].

**Opsin identification and phylogeny**. A total of 1247 opsins from all the major metazoan groups were used as seeds in a BLAST search of the predicted protein sequences of *C. dipterum*, *E. danica* (Edan_2.0; https://www.ncbi.nlm.nih.gov/assembly/GCA_000507165.2/) and *L. fulva* (Lful_2.0; https://www.ncbi.nlm.nih.gov/assembly/GCA_000376725.2/). To this set of sequences, additional mayfly LWS, UV and UV-opsin sequences from transcriptome assemblies for *Baetis sp.* EP001 and *Epeorus sp.* EP006[33,72], for *Baetis sp.* AD2013 from ref. [1], and *Baetis rhodani* shotgun whole-genome assembly[73], were obtained. Mayfly LWS, UV and Blue-opsin sequences were carefully inspected and curated (Supplementary Data 18). Phylogenetic reconstruction was performed using Ultrafast bootstrap with 1000 replicates, aLRT Bootstrap and aBayes[74,75]. In all the phylogenetic analyses the trees were rooted using the melatonin receptor which represents the opsin's closest outgroup[76] (see Supplementary Note 5 for details).

**Phylostratigraphy**. To classify genes by origin (phylostratigraphy) we used an expanded dataset of 28 species (see Supplementary Note 6). OrthoFinder2[61] was used to compute orthogroups. This resulted in 13 phylostrata for *Cloeon* (see Supplementary Data 6). The proportion of gene ages/phylostrata in a subset of interest was compared to the background proportion of ages of the species of interest using a fisher exact test fisher.test (alternative="two.sided") in R (see Supplementary Note 6).

**RNAi assays in *Drosophila* wings**. Vienna Drosophila Research Centre (VDRC) lines (see Supplementary Data 15 and Supplementary Note 7) were crossed to *yw; nubbing-Gal4 (nub-Gal4); +* line to express the RNAi constructs specifically in the wing. Crosses were kept at 25 °C for 48 h and then switched to 29 °C. Wings were dissected from adult females and mounted in Hoyer's/Lactic acid (1:1) medium prior to microscopic analysis and imaging.

**Tau index**. To determine whether genes were expressed in a tissue-specific manner or ubiquitously throughout our developmental stages and organ samples, we calculated which genes had a tau index (from 0 to 1) higher than 0.8 through the formula: $sum(1 - x/max(x))/(length(x) - 1)$, according to their cRPKM values. To calculate tau index for OBP genes, we used normalised counts obtained with samtools utilities (http://www.htslib.org/) and htseq-count[77].

**Tissue-specific transcriptomics**. We calculated which genes where expressed in wing pads and one of the other tissues preferentially, according to cRPKM. We considered that the minimum expression of the test group was 20 and that the difference between the test group (wing pad and second tissue) with the rest of the tissues was at least of 30% (Supplementary Fig. 6d, e, Supplementary Note 7).

**Reporting summary**. Further information on research design is available in the Nature Research Reporting Summary linked to this article.

## Data availability

All data generated and analysed during this study are available in European Nucleotide Archive (ENA) public repository with the project accessions PRJEB34721 and PRJEB35103. The assembly accession is GCA_902829235 (sample ID ERS4386951, contig accession CADEPI010000001-CADEPI010001395). All other RNA-Seq datasets and genome assemblies used in the study are publicly available and listed in Supplementary Data 13 and Supplementary Data 5, respectively. The assembly and annotation are also available as a UCSC track hub: http://ucsc.crg.eu/ and https://genome.ucsc.edu/s/IsabelAlmudi/Cdip_genome.

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

## Acknowledgements

We thank Yamile Márquez, Juan J. Tena, Alejandro Gil and Rafael D. Acemel for advice on bioinformatics analyses and Andrea Luchetti for providing proteome annotation of *L. arcticus*. This project was mainly funded by the European Union's Horizon 2020 research and innovation programme under the Marie Sklodowska-Curie Grant Agreement 657732 to I.A., Grant BFU2015-66040-P to F.Ca., institutional Grant MDM-2016-0687 (MINECO, Spain). Additional funding was provided by the European Research Council (ERC) under the European Union's Horizon 2020 research and innovation program (ERC-StG-LS2-637591 to M.I.), the Spanish Ministerio de Ciencia (BFU2017-89201-P to M.I., RYC-2016-20089 and PGC2018-099392-A-I00 to I.M.).

## Author contributions

T.A., M.Gu., A.dM., M.I., I.M., F.Ca. and I.A. contributed to concept and study design. J.V., A.dM., F.M., P.F., C.W., R.F., P.M., F.Cr., J.G.G., M.Gu., T.A., C.V.-C., J.G., J.P., R.L., J.R., A.S.-G., M.I., I.M., F.Ca. and I. A. performed computational analyses and data interpretation. K.D., S.A., B.M., F.Cr., J.G.G., M.Gu., T.A., M.I., F.Ca. and I.A. obtained biological material and generated next-generation sequencing data. G.M., A.A. and I.A. performed in situ hybridizations and knockdown assays. I.A. and F.Ca. coordinated the project and obtained funding. I.A. and F.Ca. wrote the main text with the help of I.M., and inputs from all authors. Arthropod illustrations by I.A., released under a Creative Commons Attribution (CC-BY) License.

## Competing interests

The authors declare no competing interests.
