## [Peer Review File · Nature Communications]

Reviewers' Comments:

Reviewer #1:

Remarks to the Author:

Almudi and colleagues report the sequencing, annotation, analysis of the de novo assembled genome of the mayfly *Cloeon dipterum*. Along with the genome, they also report analyses of gene expression in a large number of tissues during development. Finally, the authors add a section where they discovered and experimentally tested genes that play a previously unknown function in wing development, using *Drosophila* as a system for these tests.

Some of the main questions of this paper revolve around adaptation of this lineage to a double life style: aquatic juvenile phase and adult aerial/terrestrial phase.

In the genomic and transcriptomic analyses, the authors found correlation between the double life style and the expansion of certain gene families as well as the temporal gene expression. Also, they found that certain opsins were male-specific and this is consistent with males having a massive pair of extra eyes on their head that allow them to track and mate with females midair. Also, they report that a set of wing-associated genes were conserved when various winged insects were compared (flies, mayflies and centipedes).

Another interesting aspect of this paper was the result that some odorant binding protein were co-expressed with neuronal markers in the gills, allowing the authors to formulate the hypothesis that gill may have a sensory function.

Overall, the data in this paper are quite interesting and will undoubtedly open many research avenues for the community. This is both in terms of resources built and questions and hypotheses raised.

I only have a few minor comments that the authors may want to consider fixing;

- The qualifier 'very' comes back repeatedly throughout the manuscript while in no case it is necessary. I suggest to remove this word entirely from the ms.
- Line 58: replace 'along' by 'throughout' or 'during'
- Lines 66-67: '...has kept a record...' this gives the impression that the adaptations are no longer there but their record is still in the genome. Please reformulate.
- Line 84: please replace 'undergo' by 'have undergone'
- Line 91-93: please reformulate and clarify this sentence
- Line 106-110: another interesting aspect about this study, which needs to be added here, is that the authors managed to overcome the fact that mayflies are extremely short-lived as adults.
- Lines 128-121: can the authors provide more metrics here about busco analyses. Specifically how many genes are duplicated, how many are fragmented and how many are complete.
- Line 161: replace 'stimulus' by 'stimuli'
- Line 173: I would suggest the title as this "Expansion and role of of Odorant Binding Proteins during the aquatic phase"
- Line 241-248: which developmental stage was used for eye RNA seq (adult?)
- Line 303: please remove the word 'in'
- Line 326: the first sentence doesn't really mean much, please reformulate.
- Line 351: please replace 'make' with 'makes'

Reviewer #2:

Remarks to the Author:

Almudi et al. here report the first comprehensive study of a paleopteran genome and transcriptomes. Overall procedures and results of data analysis are well-presented. Through the data analysis authors successfully found characteristic gene copy number and controlled expression in both OBP and Opsin genes, which might have contributed to adaptation to distinct environments along the life cycle of this

mayfly. Remarkably, authors unveiled possible function of nymphal abdominal gills in chemoreception, which suggests a previously unexpected physiological role of gills in aquatic insects. This novel finding opened my view on how this insect senses the aquatic world during nymphal stage, and will inspire both physiologists and ecologists.

Authors also claims two points in relevance to the evolutionary origin of wings, although careful revision may be required for both points: (1) identification of the evolutionary conserved core wing genes among pterygote insects, and (2) the developmental similarity between wing and gill in Cloeon. For the first point, authors' statement "a core set of deeply conserved wing-specific genes" is not convincing to me and would be weakened because of the following reason: the transcriptome similarity as a consequence could be reasoned to either evolutionary conservation from the common ancestor (i.e. homology) or convergence. As two species used for the pairwise comparisons in this study (either two of the mayfly Cloeon, the fly *Drosophila* and the centipede *Strigamia*) are phylogenetically distant to each other, it is risky to conclude that shared gene expression is due to evolutionary conservation since the date back to the common ancestor, rather than to convergence. Authors should discuss both possible interpretations. For the second point, the transcriptome similarity between wing and gill, the result itself is not very surprising because these organs were only two appendages included for the clustering analysis. Authors should clarify the gene expression that led the clustering of wing and gill, in order to understand shared developmental procedure between two organs.

I listed specific comments to each point below:

Line 63

As I mentioned earlier, it is risky to conclude that genes authors identified are "a core set of deeply conserved wing-specific genes at the root of the pterygote insects". In addition, it is unclear whether genes that show shared expression between the wing of Cloeon and that of *Drosophila* are wing-specific or pleiotropic.

Line 73

"The Pterygote" should be replaced to "Pterygota" or "pteryogote insects".

Line 101, Fig. 1b

It would be helpful to indicate the first and second eyes with arrows in Fig. 1b.

Line 137

Which stage of nymphs were organ samples collected from? Please indicate the stages and interpret results from following analyses based on the stages. For example, wings in early and late stage nymphs should mainly undergo different developmental procedures, such as growth and venation.

Line 218

The essentiality of visual cues is not directly shown from the data in Fig. 1d. Other citations or speculations are needed here.

Line 293

Please clarify how the 8 genes were selected out of 96 genes. As all genes tested in *Drosophila* indicate some phenotypes on wing, it looks like a biased presentation without a clear explanation.

Line 318

Please list up known functions or GOs of these 42 genes as a supplementary data, and discuss what it suggests. It might be expected to obtain the clustering of wing pad and gill as these organs are only two appendages included in this comparison. It is important to discuss whether genes allowed this clustering reflect the developmental similarities between two organs, or the clustering was just

because of missing of other organs that allowed to test alternative hypothesis.

Line 408

Are there any replicate samples for RNA-seq? Please clarify it.

Figure 3

Please specify the species name of "ancestral odonata" in Figure 3a.

Reviewer #3:

Remarks to the Author:

Review of "Genomic adaptations to aquatic and aerial life in mayflies and the origin of 2 wings in insects" by Almudi et Al.

Almudi et al provide a comprehensive genomic investigation of a mayfly (*Cloeon dipterum*)– a critical taxonomic group to understand the evolution of winged insects. This particular species genome will be well referenced because of it's phylogenetic position and interesting biology.

The Draft genome assembly appears to be done well (the contig N50 is excellent), but is supplemented and improved with a comprehensive transcription analyses, which will be well received and necessary for this group.

Specific nice results are:

Odorant proteins are expanded – something know in general for insects, but nice to see in this group – but more importantly the gill expression in nypal stages is a great observation that will help shed light on the different life stages beyond the usual adult only studies done in *Drosophila*, *Mosquitos* etc.

The visual biology story is nice with the sexually dimorphic opsin.

Refining the core set of wing genes is particularly nice, especially with the *Drosophila* functional followup. The transcriptional similarities between wings and gills, is also extremely interesting for how these critical wing structures evolved, - quite an amazing result, and it suggests questions (for other groups) about what these expression networks can do in other taxa such as crustacea?

Overall this is nice work performed very well, in a critical taxa for all entomologists and I suggest rapid publication.

Minor Issues.

1. I can't find the sequences online at ENA – I presume they have been held until publication – but please release them and have the editor confirm that before publication. – Can I suggest you do this pre-publication in the future – especially if the pre-print is available. It is more likely to get you collaborators that for someone to try and scoop you in todays world. Alternatively I missed them?

2. There are a few web links to an individual lab for a couple of species: - could you please change the reference to either a publication or NCBI for a long-term repository to avoid link rot.

Ephemera danica (<https://www.hgsc.bcm.edu/arthropods/mayfly-genome-project>) and *Ladona fulva* (<https://www.hgsc.bcm.edu/arthropods/scarce-chaser-genome-project>).

3. A comment, For the future (But NOT necessary for this publication in anyway) it would be nice to add hiC Scaffolding to the genome assembly to put the sequence in a chromosomal context. – The contigs are great, so that would make it even better.

Reviewer #4:

Remarks to the Author:

Almudi et al. report results from the sequencing, assembly, and annotation of the genome of the cosmopolitan mayfly *Cloeon dipterum*, together with extensive transcriptomic sampling of several life stages and tissues. Analyses of these data revealed (i) an expanded repertoire of genes encoding odorant binding proteins, many of which were expressed in nymph gills, confirmed with in situ hybridization; (ii) duplicated opsin-encoding genes, with expression in adult male-only turbanate eyes, confirmed with in situ hybridization. Comparisons with *Drosophila melanogaster* expression data revealed (iii) conserved gene expression modules associated with gut, muscle, brain, and Malpighian tubules (many of which were also found in comparisons with the centipede, *Strigamia maritima*); (iv) conserved gene expression modules associated with wings and wing development, testing with RNAi in *Drosophila* of eight of these genes for which no wing-related role had previously been reported all produced clear wing phenotypes. Comparing life-stage and tissue gene expression modules revealed that (v) gills were the most closely related organ to developing wings, in terms of genes in common. The authors conclude (a) sensory specialization means adaptation of nymphs to rely on chemical stimuli processed by the gills while adults depend more on vision; (b) core gene expression modules are conserved in different phyla; (c) a core wing development expression module is conserved in pterygote insects and shares a common genetic programme with gills. The study is elegant and the results are enlightening, but to be fully convinced I would like to see evidence that the results are also robust.

Key criteria.

The data is technically sound: yes, I believe that the data are technically sound.

The paper provides strong evidence for its conclusions: yes, the authors provide strong evidence for their conclusions, but see points below.

The results are novel: yes, I believe the results are novel.

The manuscript is important to scientists in the specific field: yes, the findings are important.

Major points.

[1] Clustering robustness

The authors use two methods for gene expression clustering: Mfuzz for defining clusters of co-regulated genes during development through embryonic and nymphal stages to the adult; and WGCNA to define modules of co-regulated genes across tissues.

Firstly, neither of these approaches is described in sufficient detail to be able to gauge their validity or to reproduce the analyses. All clustering methods are sensitive, to varying degrees, to the parameters and cut-offs used as well as to the filtering and normalisation steps employed. It is therefore critical to detail each step of these two clustering analyses in order to have confidence in the resulting Mfuzz clusters and WGCNA modules.

This is doubly important because the majority of the key results depend on the gene membership of each cluster or module. If gene membership is too unstable then almost any minor tweak to the analysis could lead to dramatically different results. Thus, as well as clearly detailing all steps and parameters etc. I would like to see evidence that the resulting clusters and modules are robust. The Mfuzz publication states "Variation of the FCM parameter m also allows investigation of the stability of clusters. We define stable clusters as clusters that show only minor changes in their structure with variation of the parameter m . Stable clusters are generally isolated and compact. This is contrasted by weak clusters that lose their internal structure or disappear if m was increased." This suggests a possible approach to gauge cluster stability. Hopefully the main clusters of interest with respect to the key results will prove to have good stability.

Additionally, if I understand correctly each gene is assigned to each of the clusters with a score between 0 and 1 that sum to 1 (I believe that the colours in Figure S2 are designed to show these scores with high-scoring pink to low-scoring green and yellow genes). Cluster membership is then defined simply by the highest score (this assignment is not actually detailed in the supplement as far as I could tell, so this is my assumption). Thus by eye in Figure S2 we can get a sense of which clusters have mostly strong membership (mostly pink) and which are more diffuse (more purple, blue, and green). Median score might therefore be a good numerical proxy for gauging the relative robustness of each cluster, e.g. it would clearly show that for clusters 12 and 18 presented in Figure 2b, cluster 12 scores are lower than cluster 18 scores. Note that when it comes to then test these gene lists for each cluster for Gene Ontology (GO) term enrichments these scores are ignored (see point [2] below for suggestions with respect to this).

The developers of WGCNA themselves stress the need to assess whether modules are preserved and reproducible (Is my network module preserved and reproducible? Langfelder et al. 2011 <https://doi.org/10.1371/journal.pcbi.1001057>). The focus of these discussions was actually on biological reproducibility between species rather than technical, but the approaches should still apply. They should also be useful for the mayfly-fly and mayfly-centipede comparisons as an alternative to the applied cross-tabulation approach (hypergeometric tests on overlaps between homologous groups for each pair of modules) for measuring module preservation without requiring module detection in the other species. Calculation of module preservation statistics is already part of WGCNA (modulePreservation function), so applying this type of analysis could provide additional evidence of cross-species module preservation. One possibility to confirm the robustness of the WGCNA modules themselves could be to apply an iterative approach e.g. iterativeWGCNA (<https://www.biorxiv.org/content/10.1101/234062v1>).

[2] gene ontology assignment and enrichment

The qualitative descriptions of the functional roles of various clusters and modules come from the tissues and/or life-stages where they are expressed combined with hints from Gene Ontology terms and PFAM domain annotations. These are clearly key to many of the results presented, but I could not find details of how GO terms were assigned, and PFAM seems to cover only a quarter of all Cdipt genes. For example, the 33-page WGCNA summary PDF simply presents each module with expression across the samples, topGO results, and the 'name' assigned, for Cdipt, Dmela, and Smari. I could not find any details about where the GO-term annotations comes from. I presume Dmela would be sourced from the corresponding gene set version at FlyBase, but how were Cdipt and Smari genes assigned GO terms? Additionally, there are no details presented about the actual tests run using topGO (which test statistics and algorithms – classic, elim, weight, parentChild, fisher, ks, etc.), and how the background was controlled – because only expressed genes can be assigned to a module it means that the background can only contain expressed genes not all genes.

Methods for GO-term enrichments of Mfuzz clusters are described in the supplement with one sentence: "Taking the *Drosophila melanogaster* orthologs, we performed Gene Ontology (GO) enrichment analysis for each of the clusters using DAVID". Given how the GO-term enrichments form such an important part of the results, helping to define the main functions of various clusters, this description is woefully inadequate. (1) While using *Drosophila* orthologues to get at function is fine I would also like to see evidence that similar terms can be identified when using Cdipt gene annotations directly because there will be many cases where orthology is not one-to-one. (2) There is no mention of the background gene set, was this correctly controlled? The foreground, for each cluster, is a list of *Drosophila* genes with orthologues in Cdipt that were expressed in Cdipt. The background therefore should not be all *Drosophila* genes provided by DAVID, but rather all *Drosophila* genes with orthologues in Cdipt that were expressed in Cdipt.

For confirming GO-term enrichments for Mfuzz clusters: currently I believe all cluster members are treated as the foreground for enrichment testing, I would suggest (1) eliminate low-scoring genes from the list before testing (but this annoyingly involves selecting a cut-off, and applying the same

filter to the background gene list); alternatively (2) using a score-based enrichment test such as the Kolmogorov-Smirnov test implemented in the topGO R package would remove the need to select a cut-off (also known as gene set enrichment analysis, GSEA). Note that the full background gene list should not be all *Cdpt* genes, but it should consist of all genes assigned to any of the clusters (this was not detailed in the methods so I cannot assess whether the background was correctly controlled or not in the current analyses). If the GO terms shown in Figure 1d are still significant then these are robust results.

Minor points.

Please note, not all minor points are listed here. I provide a scanned copy of my comments on the main text, and I also provide a copy of the supplement with tracked-changes and further comments for consideration.

Abstract: while I realise it is difficult to get all major results to fit into the abstract, I feel that the gill-wing result should appear in the abstract.

Please make sure all acronyms are defined on their first usage, this is very important for readability, e.g. Myr; CEGMA; BUSCO; HRP; WGCNA; ONT; HMM; DAPI; etc.

Please make sure all panels of all figures are correctly referenced from the main text, e.g. Figure 2b and 4b do not appear to be referenced in the text. Also, the use of primes, e.g. a and a', to distinguish some panels is hard to see – please use distinct letters instead.

Introduction: while I appreciate the brevity as it stands now, it would seem appropriate to mention current thinking on the origin of wings somewhere in the introduction.

Figure 1: the title does not reflect the content – what does the figure show about the genome?

L123-127: genome size estimates should probably also be mentioned in the main text, not just the supplement. In the supplement clearer details on how k-mers were chosen need to be provided to make sense of the analyses performed.

L128: was assembled in 1395 scaffolds, with N50 of 0.461 Mb. But please check because in SOM it says CONTIG N50 = 461,411 bp and SCAFFOLD N50 = 434,950 bp. This is strange – how can scaffold N50 be shorter than contig N50, please double check.

L130: is the CEGMA assessment of the gene set or of the genome assembly? Same for BUSCO, the text suggests the gene set was assessed, but the supplement describes the assembly assessment, what were the results for the genome assembly and what were the results for the gene annotation – please clarify exactly what is being presented. Also, please present full BUSCO results, i.e. include duplicated and fragmented percentages as this is needed to give a full picture. E.g. if % duplicated is alarmingly high then the assembly could have failed to collapse haplotypes, which would then raise questions about the reported result of the large number of OBPs that were identified. For example, the opsins section indicates that “opsin sequences were carefully inspected and curated to detect gene annotation errors such ... and identical sequences derived from haplotypes of the same gene” which suggests that there were some ‘haplotypes’ – is this an issue or not, please address.

L133: In Table S4 please give actual versions of annotation sets used, not just source DB, e.g. *Caenorhabditis elegans* EnsEMBL; *Heliconius melpomene* EnsEMBL; *Ixodes scapularis* Uniprot; etc.

are not useful for replication.

L133-135: it appears strange to mention gene gain and loss here but not present it as a result per se "allowing the study of gene gains and losses at the origin of winged insects" is all we get here, and then all of a sudden much later in lines 309-311 there is a return to the gain/loss results. If there is a result to present for Figure 1a then present it properly.

Methods to infer gene gains and losses are not clearly described. The supplement simply points to a reference that seems to be about opsins. The section on gains/losses ends with "The *C. dipterum* Pfam and GO annotations were used to inspect the functions of these genes." Where did the GO annotations come from?

L137-142: for transparency please clearly indicate in the main text which samples had replicates and which did not, which were PE and which were SE. This is given in the supplement but it currently feels hidden beyond merely for the sake of brevity. There were a couple of places where I wondered whether the lack of a replicate for E6 would have an impact on the type of analysis that could be performed – please indicate clearly where E6 is included or excluded because of having only one replicate and openly describe this (and the single replicates for nymphal heads) in the supplement. Also why was Female Head replicate 2 excluded from Figure 4e?

Particularly for readers interested in technical aspects, please add a brief mention in the main text and some commentary to the supplement on why, given 96x Illumina coverage and 36x Nanopore coverage (with N50 3.5 Kbp) and inbreeding (e.g. did this work?), the contiguity of the resulting hybrid assembly was not really outstanding: ~1400 scaffolds N50 ~460 Kbp. E.g. looking at scaffold-level assemblies at NCBI of similar sizes (150-210 Mbp) with similar number of scaffolds (1000-3000), which so far as I can tell did not have any long reads, all have higher N50s: *Fopius arisanus* 978,588; *Polistes dominula* 1,625,592; *Orussus abietinus* 612,083; *Drosophila obscura* 472,512; *Cephus cinctus* 622,163; *Galendromus occidentalis* 896,831; *Drosophila pseudoobscura* 1,459,550.

Annotation: from supplement "... Augustus gene prediction tool which yielded 16,364 evidence-based gene models ... These models contain 4,308 non-redundant PFAM models as assigned using the PfamScan tool" – that is just 26%, is this normal/expected? Would InterProScan give better domain assignment coverage than just PFAM? Also, transcriptome mapping identified about double the number of putative gene loci, so why was the final prediction so much lower?

Mfuzz vs. WGCNA: as some readers will not be too familiar with such gene expression clustering methods, for clarity please briefly indicate in simple language why Mfuzz was used for time-course gene expression clustering and WGCNA was used for delineating modules of co-regulated genes across tissues. This should be then elaborated on in the corresponding supplementary text sections, which are currently also severely lacking in detail. For example, a key feature of Mfuzz is that genes can belong to more than one cluster – not all readers will be aware of this and it is an important fact when trying to interpret the figures and main results. I'm guessing here that the best-scoring cluster was used to define membership, but this is not detailed as far as I could see.

Mfuzz methods: the level of detail given is totally inadequate. What fuzzifier 'm' value was used? And how was it selected? What data exactly were used? Counts? Were they log-transformed? E.g. Mfuzz website says "Since it is common to log-transform the FPKM before clustering (although it is not absolutely required) ..." Was Mfuzz standardisation also carried out? And what about low/no expression genes? E.g. were genes with low/no values in most samples first filtered out? How were 'missing values' dealt with? How exactly was the optimum number of clusters decided? Then to finish, what do the results actually look like? E.g. How many genes, and what level of overlap was there

amongst the resulting clusters? Table S8 seems to indicate that Cdip genes each belong to only a single cluster – how was this decided, and what cut-off was applied for a gene to belong to a cluster or not? What proportion of cluster members are considered core members, e.g. applying alpha core cut-off of 0.7? How can cluster robustness be demonstrated?

L151: here and elsewhere, please be very careful when using the term 'specifically' or 'specific' when referring to expression patterns. Normally 'specific' expression of certain genes suggests that these genes are not at all expressed anywhere else. Please rephrase here and elsewhere to be clear about exactly what is meant when describing these expression patterns.

L173: proteins do not expand, families expand, please rephrase. Also L222 and elsewhere.

For gene family and orthology assignments, if some of the species do not yet have a 'genome publication' to reference then it would be appropriate to acknowledge people or consortia who granted permission to use such pre-publication data in the main acknowledgements section.

L181-189: please be honest about the numbers of complete gene models identified, there are several discrepancies between what is reported here in the main text and what is presented in the supplement. Also, the gene annotation pipeline produced alternative transcripts, yet for these gene families of biological interest no mention is made of whether the protein repertoire is actually larger than the gene repertoire as would be the case through alternative splicing.

L191: the use of the word 'previous' here and elsewhere, suggests results from a previous published study please rephrase to be clear about exactly what it meant. Also L304.

L199: nymph-specific, again, careful with the use of 'specific', especially here because clearly clusters 18 and 12 show first increase in expression already in E14, i.e. technically before the nymph stage.

L206-207: 34% - please check, this seemed confusing with respect to the numbers presented in the sentence. Also, I could not find the methods for building the heatmap shown in 2c.

L235-240: 'expansion' why not just state that there are 4 copies?

L246: "whose protein sequence is highly modified" how does either 3b or 3c show how the protein sequence is modified?

WGCNA methods: The level of detail given is totally inadequate. WGCNA recommends log-transforming RPKMs – was this done? WGCNA website "We do not recommend attempting WGCNA on a data set consisting of fewer than 15 samples." 14 for Strigamia is only just below their recommendation, so it might be alright, but this should surely be mentioned. The sample information in Table S11 is confusing: Sample counts do not correspond to sample lists. Cdip samples: 26, but 27 in list (one female head excluded – why, because no replicate, but E6 is in the list of samples?) Dmel samples: 19, but 16 in the list, e.g. Adult_Fem_Heads and Adult_Male_Heads and Embr_12_14h and others are not in the list Smar samples: 14, but 7 in the list. How were all these samples chosen and what effect does the choice of sample have on the resulting modules? I.e. if you change the input data do you still get to see the same major modules appearing or is this highly sensitive to the data? Filtering: "We performed the analyses using as datasets genes that were present in a least two species in our family reconstructions and showed variance across samples" – if I understand this correctly it means that any genes in Dmela, Cdipt, and Smari, that do not have orthologues in at least one of the other species would be filtered out, correct? Please explain the reasoning, and what 'family reconstructions' are – OrthoFinder with 14 species or 28 species?

L272: the methods for these tests are not described in sufficient detail to assess or reproduce.

L277: clustering methods that produce the dendrograms and hence determine the orders of tissues and life-stages presented are not explained. Fig 4a Malpighian tubule is incorrectly shown as tube. There is more here to discuss: testis-testis is weak, but present; ovaries-ovaries are nice and strong; head/vision seems good; but how to explain ovaries-wing discs and testis fat body? (also methods for Fig2c clustering are not described).

L284: what exactly does 'shared' mean here? Genes in the modules that have orthologues? And if 130 are shared, is that 130 Drosophila genes or 130 Cloeon genes? And this is out of how many genes in each of the modules?

L305: where is this result shown? Also, where are the methods that describe exactly how tau indices were calculated?

L316: 'clustering' method seems not to be described anywhere (like other heatmaps), yet the fact that these two modules cluster together is being presented as a key result – methods must be detailed.

L319: the overlap analysis presented at the end of the supplement is very brief, please make sure this is fully described, and confirm that 42/98 is statistically significant.

L355-356: 'very specific genomic changes' This is very vague, and what exactly is the discussion point here? Probably just remove this sentence.

L409: please clarify sample counts – table seems to give 37, 13 in 2x replicate, 1 with no replicate, then 10 heads.

Supplementary tables: the numbering has gone awry with these tables, and the text refers to Table S17 whilst there are only 15 supplementary tables. To reduce the number of supplementary tables that are currently separate files and require some back-and-forth for the reader to cross reference text and table data, I would suggest that smaller tables e.g. those that would fit on a single page, be incorporated into the main supplementary materials document.

Methods: the level of detail provided in the main text is extremely variable, e.g. opsin identification and analysis gets a rather detailed summary while other key methodological aspects are completely ignored such as GO term assignment and enrichment analyses. Please harmonise the methods summary descriptions in the main text so that nothing is ignored, and include explicit references to the further details provided in the supplement.

Reviewers' comments:

Reviewer #1 (Remarks to the Author):

Almudi and colleagues report the sequencing, annotation, analysis of the de novo assembled genome of the mayfly *Cloeon dipterum*. Along with the genome, they also report analyses of gene expression in a large number of tissues during development. Finally, the authors add a section where they discovered and experimentally tested genes that play a previously unknown function in wing development, using *Drosophila* as a system for these tests.

Some of the main questions of this paper revolve around adaptation of this lineage to a double life style: aquatic juvenile phase and adult aerial/terrestrial phase.

In the genomic and transcriptomic analyses, the authors found correlation between the double life style and the expansion of certain gene families as well as the temporal gene expression. Also, they found that certain opsins were male-specific and this is consistent with males having a massive pair of extra eyes on their head that allow them to track and mate with females midair. Also, they report that a set of wing-associated genes were conserved when various winged insects were compared (flies, mayflies and centipedes).

Another interesting aspect of this paper was the result that some odorant binding protein were co-expressed with neuronal markers in the gills, allowing the authors to formulate the hypothesis that gill may have a sensory function.

Overall, the data in this paper are quite interesting and will undoubtedly open many research avenues for the community. This is both in terms of resources built and questions and hypotheses raised.

AR: We thank the Reviewer for his/her positive and constructive comments.

I only have a few minor comments that the authors may want to consider fixing:

- The qualifier 'very' comes back repeatedly throughout the manuscript while in no case it is necessary. I suggest to remove this word entirely from the ms.

AR: We removed the qualifier 'very' from the manuscript, except in Line 110: "*...overcoming past challenges to study paleopterans which are generally not very amenable to rear in the lab.*"

- Line 58: replace 'along' by 'throughout' or 'during'

AR: Replaced by 'throughout'

- Lines 66-67: '...has kept a record...' this gives the impression that the adaptations are no longer there but their record is still in the genome. Please reformulate.

AR: We modified the sentence as "*... paleopteran insect has uncovered genetic basis of key evolutionary adaptations in mayflies and winged insects.*"

- Line 84: please replace 'undergo' by 'have undergone'

AR: It has been replaced accordingly

- Line 91-93: please reformulate and clarify this sentence

AR: We reformulated the sentence as follows: "evolutionary changes that should be mirrored by modifications in their genomes"

- Line 106-110: another interesting aspect about this study, which needs to be added here, is that the authors managed to overcome the fact that mayflies are extremely short-lived as adults.

AR: We have added in the text: "... laboratory; with mayflies having the additional challenge of being extremely short-lived as adults"

- Lines 128-121: can the authors provide more metrics here about busco analyses. Specifically, how many genes are duplicated, how many are fragmented and how many are complete.

AR: We included the percentages of different BUSCO classes in the supplementary material and main text. The results of BUSCO (run in mode: genome; BUSCO version is: 3.0.2) for *C. dipterum* assembly are:

C: 96.9% [S:94.1%, D:2.8%], F:1.3%, M:1.8%,n:1658

1607	Complete BUSCOs (C)
1561	Complete and single-copy BUSCOs (S)
46	Complete and duplicated BUSCOs (D)
22	Fragmented BUSCOs (F)
29	Missing BUSCOs (M)
1658	Total BUSCO groups searched

We added to Supplementary Table 1 (Genome Statistics) new sheets with BUSCO information and modified the main text to provide some metrics: " The gene completeness of the genome was estimated to be 96.77% and 98.2%, according to *Core Eukaryotic Genes Mapping Approach* (CEGMA v2.5 ¹⁶) and *Benchmarking Universal Single-Copy Orthologs* (BUSCO v3 ¹⁷) respectively. Moreover, the fact that we have found 94.1% Complete Single-copy and only 2.8% Complete Duplicated BUSCOs, out of 1,658 insect orthologues (See Supplementary Information), supports the haploid nature of our reference genome."

- Line 161: replace 'stimulus' by 'stimuli'

AR: It has been replaced accordingly

- Line 173: I would suggest the title as this "Expansion and role of Odorant Binding Proteins during the aquatic phase"

AR: We have modified the title to "Expansion and role of Odorant Binding Protein genes during the aquatic phase", to take into account the comments of the two Reviewers.

- Line 241-248: which developmental stage was used for eye RNA seq (adult?)

AR: We added to the sentence: "...in adult male heads..." Moreover, Supplementary Table 6 has been expanded to incorporate the stages in which the samples were obtained.

- Line 303: please remove the word 'in'

AR: Done

- Line 326: the first sentence doesn't really mean much, please reformulate.

AR: We eliminated the sentence.

- Line 351: please replace 'make' with 'makes'

AR: Replaced

Reviewer #2 (Remarks to the Author):

Almudi et al. here report the first comprehensive study of a paleopteran genome and transcriptomes. Overall procedures and results of data analysis are well-presented. Through the data analysis authors successfully found characteristic gene copy number and controlled expression in both OBP and Opsin genes, which might have contributed to adaptation to distinct environments along the life cycle of this mayfly. Remarkably, authors unveiled possible function of nymphal abdominal gills in chemoreception, which suggests a previously unexpected physiological role of gills in aquatic insects. This novel finding opened my view on how this insect senses the aquatic world during nymphal stage, and will inspire both physiologists and ecologists.

AR: We are grateful for the Reviewer positive comments and suggestions

Authors also claims two points in relevance to the evolutionary origin of wings, although careful revision may be required for both points:

(1) identification of the evolutionary conserved core wing genes among pterygote insects, and (2) the developmental similarity between wing and gill in Cloeon.

For the first point, authors' statement "a core set of deeply conserved wing-specific genes" is not convincing to me and would be weakened because of the following reason: the transcriptome similarity as a consequence could be reasoned to either evolutionary conservation from the common ancestor (i.e. homology) or convergence. As two species used for the pairwise comparisons in this study (either two of the mayfly Cloeon, the fly *Drosophila* and the centipede *Strigamia*) are phylogenetically distant to each other, it is risky to conclude that shared gene expression is due to evolutionary conservation since the date back to the common ancestor, rather than to convergence. Authors should discuss both possible interpretations.

AR: Evolutionary convergence is indeed a very important issue when studying conservation of gene expression, especially in studies using candidate gene approaches where only a limited and pre-selected number of genes are compared between species. In the case of unbiased genome-wide comparisons such as the WGCNA module comparison included in our study, we think it is unlikely that the statistically significant similarities in module composition we observed in homologous tissues (including wings) between *C. dipterum* and *Drosophila* using two different methods (see new version of Supplementary Figure 6) are due to evolutionary convergence rather than conservation. However, we cannot completely rule out that a few particular genes among the 126 orthologous gene families found both in wing pad and wing disc modules were independently recruited to wings in the two species. But, this type of similarities due to convergence are expected to occur between any two random modules and not just in the wing modules. Thus, we could expect that any pair of modules could have a few shared genes merely by chance. But, if convergence were rampant, we would have observed many statistically significant similarities between unrelated modules, which is not the case.

Nevertheless, we think that taking into account evolutionary convergence is always important. Thus, we have toned down the manuscript accordingly and we now explicitly discuss this possibility in the discussion, indicating that further studies in additional species and developmental stages will be necessary to refine this first list of shared genes and to identify potential cases of convergence.

For the second point, the transcriptome similarity between wing and gill, the result itself is not very surprising because these organs were only two appendages included for the clustering analysis. Authors should clarify the gene expression that led the clustering of wing and gill, in order to understand shared developmental procedure between two organs.

AR: We agree with the Reviewer that wings and gills are the only appendages in this study and this could have had an effect in the clustering. Nevertheless, we considered that having these RNA samples was a great opportunity to make a first transcriptomic approximation to some of the hypotheses proposed regarding the origin of wings and in particular, the role that abdominal gills of Ephemeroptera could have had in this process. In fact, when we checked the 42 genes that are highly expressed in wings and gills of the mayfly and searched for the functions of their *Drosophila* orthologs (now included in the main text and a new Supplementary Table), we found that 40% of them have described roles in wing development, supporting an actual similarity of these transcriptomic programs. Nonetheless, as the Reviewer points out, a more detailed analysis will be necessary in the future to fully conclude that wings originated from gills in the first winged insects. Thus, we have included in the discussion some clarification of our results and conclusions to make clear the potential limitations of this comparison.

I listed specific comments to each point below:

Line 63

As I mentioned earlier, it is risky to conclude that genes authors identified are “a core set of deeply conserved wing-specific genes at the root of the pterygote insects”. In addition, it is unclear whether genes that show shared expression between the wing of *Cloeon* and that of *Drosophila* are wing-specific or pleiotropic.

AR: As commented above, we have now toned down the manuscript, stating that although conservation is the most likely scenario for most of these genes, there could be cases of independent recruitment and that it will be necessary to study additional species and developmental stages. With regards to the specificity, these genes were assigned to wing modules in the WGCNA analyses, showing higher expression in the wings than in the other samples, but this does not imply that they could not have additional functions in other tissues and processes, therefore we have replaced “specific” by “associated”.

Line 73

“The Pterygote” should be replaced to “Pterygota” or “pterygote insects”.

AR: It has been replaced by “Pterygota”

Line 101, Fig. 1b

It would be helpful to indicate the first and second eyes with arrows in Fig. 1b.

AR: We have added arrows and arrowheads in panels 1b” and 1b''' (now renamed as 1d and 1e) pointing to the compound and turbanate eyes, respectively.

Line 137

Which stage of nymphs were organ samples collected from? Please indicate the stages and interpret results from following analyses based on the stages. For example, wings in early and late stage nymphs should mainly undergo different developmental procedures, such as growth and venation.

AR: We included an additional column in Supplementary Table 6 with details about RNA samples. We agree with the Reviewer on the importance of the stage in which wings were collected. Our wing samples were collected from late nymphal individuals, therefore, developmental processes related to ‘late wing specification’ such as venation, were most likely to be observed at that stage instead of processes related to epithelium growth or proliferation and in fact, this is what we found; the phenotypes we obtained in our knock down experiments in *D. melanogaster* wings were mostly related to venation defects rather than defects in wing size. We modify the following sentence to highlight this: “...anatomical wing features, consistent with the specificity of these transcriptional programs *and the late nymphal stages of our wing pad RNA samples.*”

Line 218

The essentiality of visual cues is not directly shown from the data in Fig. 1d. Other citations or speculations are needed here.

AR: We reformulated the sentence to make clear that we are referring to the results we obtained by Mfuzz analyses, described in the previous section: " Gene expression dynamics and GO enrichment analyses showed that visual perception must play a prominent role during adulthood in mayflies"

Line 293

Please clarify how the 8 genes were selected out of 96 genes. As all genes tested in *Drosophila* indicate some phenotypes on wing, it looks like a biased presentation without a clear explanation.

AR: We clarified in the Supplementary Information how the 8 genes were selected: "We selected eight genes of the 130 shared orthologs between the wing disc and wing pad modules based on the following criteria: availability of RNAi line in a stock centre, lack of off-target effects and viability of such line, reported undetermined function in the wing in Flybase and presence of orthologs in other species to focus on genes that appeared near the origin of pterygote insects (absent in non-winged insects and present in winged insects or genes that arose by gene duplications at the base of pterygotes)"

Line 318

Please list up known functions or GOs of these 42 genes as a supplementary data, and discuss what it suggests. It might be expected to obtain the clustering of wing pad and gill as these organs are only two appendages included in this comparison. It is important to discuss whether genes allowed this clustering reflect the developmental similarities between two organs, or the clustering was just because of missing of other organs that allowed to test alternative hypothesis.

AR: We have included a new 'Supplementary Table 15' where we listed the 42 genes that are highly expressed in wings and gills and information about their *Drosophila* orthologs, links to Flybase database, whether they have wing phenotypes described (based on Flybase information) and associated GO terms. Although we agree with the Reviewer that it will be necessary a broader RNA dataset, - with more appendages and temporal series- we thought that it was worthy to get a first transcriptomic approximation to this topic. Indeed, checking those 42 genes and what it is known about their *Drosophila* orthologous (30 out of these 42), we found that a substantial proportion of them, 40% (12/30) have wing phenotypes or described functions in wing development. This suggests that these similarities (or clustering) between these two tissues do not arise simply due to the lack of other appendage samples in the analysis but that they probably have transcriptomic programs in common.

We have included the following sentences to the main text: "Moreover, when checking the function of these *Drosophila* orthologs, 40% (12/30) of them had a known role in wing development (e. g., *shifted* and *taxi*)." and "Here, we have uncovered transcriptomic similarities between gills and wings. As the first genome-wide expression comparison of these mayfly organs, further work including additional appendages and developmental stages as well as additional mayfly species will be needed to better understand the biological meaning of these similarities. Regardless, our results would be consistent ..."

Line 408

Are there any replicate samples for RNA-seq? Please clarify it.

AR: We have included now in the main text (figure legend) and in methods section the number of replicates per RNA-seq sample.

Figure 3

Please specify the species name of “ancestral odonata” in Figure 3a.

AR: What we meant here was the opsin complement inferred in Suvorov et al 2017, for the last common ancestor of Odonata. So, we did not refer to any individual species, but to their common ancestor. We included in the figure legend “(‘ancestral odonata’ refers to the Opsin complement of the last common ancestor of this lineage as inferred in 71)”

Reviewer #3 (Remarks to the Author):

Review of “Genomic adaptations to aquatic and aerial life in mayflies and the origin of wings in insects” by Almudi et al.

Almudi et al provide a comprehensive genomic investigation of a mayfly (*Cloeon dipterum*)— a critical taxonomic group to understand the evolution of winged insects. This particular species genome will be well referenced because of its phylogenetic position and interesting biology.

The Draft genome assembly appears to be done well (the contig N50 is excellent), but is supplemented and improved with a comprehensive transcription analyses, which will be well received and necessary for this group.

Specific nice results are:

Odorant proteins are expanded – something known in general for insects, but nice to see in this group – but more importantly the gill expression in nymphal stages is a great observation that will help shed light on the different life stages beyond the usual adult only studies done in *Drosophila*, Mosquitos etc.

The visual biology story is nice with the sexually dimorphic opsin.

Refining the core set of wing genes is particularly nice, especially with the *Drosophila* functional follow up. The transcriptional similarities between wings and gills, is also extremely interesting for how these critical wing structures evolved, - quite an amazing result, and it suggests questions (for other groups) about what these expression networks can do in other taxa such as Crustacea?

Overall this is nice work performed very well, in a critical taxa for all entomologists and I suggest rapid publication.

AR: We are very thankful for the very positive comments and recommendations of the Reviewer

Minor Issues.

1. I can't find the sequences online at ENA – I presume they have been held until publication – but please release them and have the editor confirm that before publication. – Can I suggest you do this pre-publication in the future – especially if the pre-print is available. It is more likely to get you collaborators that for someone to try and scoop you in today's world. Alternatively I missed them?

AR: The Reviewer is right. The sequences were not made available, they are now. We apologise for this inconvenience and we will follow the Reviewer advice in the future and will make the sequences public as soon as they are uploaded.

2. There are a few web links to an individual lab for a couple of species: - could you please change the reference to either a publication or NCBI for a long-term repository to avoid link

rot. *Ephemera danica* (<https://www.hgsc.bcm.edu/arthropods/mayfly-genome-project>) and *Ladona fulva* (<https://www.hgsc.bcm.edu/arthropods/scarce-chaser-genome-project>).

AR: These web links have been replaced by NCBI links:

“*C. dipterum*, *E. danica* (Edan_2.0;

https://www.ncbi.nlm.nih.gov/assembly/GCA_000507165.2/) and *L. fulva* (Lful_2.0;

https://www.ncbi.nlm.nih.gov/assembly/GCA_000376725.2/).”

3. A comment, for the future (But NOT necessary for this publication in anyway) it would be nice to add hiC Scaffolding to the genome assembly to put the sequence in a chromosomal context. – The contigs are great, so that would make it even better.

AR: This is a great suggestion that we will take into consideration for the future. We agree with the Reviewer that a chromosome grade genome will be extremely useful to investigate long range regulation and chromatin interactions in this species.

Reviewer #4 (Remarks to the Author):

Almudi et al. report results from the sequencing, assembly, and annotation of the genome of the cosmopolitan mayfly *Cloeon dipterum*, together with extensive transcriptomic sampling of several life stages and tissues. Analyses of these data revealed (i) an expanded repertoire of genes encoding odorant binding proteins, many of which were expressed in nymph gills, confirmed with in situ hybridization; (ii) duplicated opsin-encoding genes, with expression in adult male-only turbanate eyes, confirmed with in situ hybridization. Comparisons with *Drosophila melanogaster* expression data revealed (iii) conserved gene expression modules associated with gut, muscle, brain, and Malpighian tubules (many of which were also found in comparisons with the centipede, *Strigamia maritima*); (iv) conserved gene expression modules associated with wings and wing development, testing with RNAi in *Drosophila* of eight of these genes for which no wing-related role had previously been reported all produced clear wing phenotypes. Comparing life-stage and tissue gene expression modules revealed that (v) gills were the most closely related organ to developing wings, in terms of genes in common. The authors conclude (a) sensory specialization means adaptation of nymphs to rely on chemical stimuli processed by the gills while adults depend more on vision; (b) core gene expression modules are conserved in different phyla; (c) a core wing development expression module is conserved in pterygote insects and shares a common genetic programme with gills. The study is elegant and the results are enlightening, but to be fully convinced I would like to see evidence that the results are also robust.

Key criteria.

The data is technically sound: yes, I believe that the data are technically sound.

The paper provides strong evidence for its conclusions: yes, the authors provide strong evidence for their conclusions, but see points below.

The results are novel: yes, I believe the results are novel.

The manuscript is important to scientists in the specific field: yes, the findings are important.

AR: We are thankful to the Reviewer for appreciating the novelty of our work and his/her thorough revision on technical and methodological aspects which has improved our manuscript.

Major points.

[1] Clustering robustness

The authors use two methods for gene expression clustering: Mfuzz for defining clusters of co-regulated genes during development through embryonic and nymphal stages to the adult; and WGCNA to define modules of co-regulated genes across tissues.

Firstly, neither of these approaches is described in sufficient detail to be able to gauge their validity or to reproduce the analyses. All clustering methods are sensitive, to varying degrees, to the parameters and cut-offs used as well as to the filtering and normalisation steps employed. It is therefore critical to detail each step of these two clustering analyses in order to have confidence in the resulting Mfuzz clusters and WGCNA modules. This is doubly important because the majority of the key results depend on the gene membership of each cluster or module. If gene membership is too unstable then almost any minor tweak to the analysis could lead to dramatically different results. Thus, as well as clearly detailing all steps and parameters etc. I would like to see evidence that the resulting clusters and modules are robust.

AR: We apologize if the Mfuzz and WGCNA analyses were not explained in sufficient detail. We have now expanded these sections in the corresponding Supplementary Information, providing all the necessary information to reproduce and validate our results, including all the relevant parameters, cut-offs and filtering and normalisations. Furthermore, we have performed additional analyses that show that our clusters are robust (see additional comments below).

The Mfuzz publication states “Variation of the FCM parameter m also allows investigation of the stability of clusters. We define stable clusters as clusters that show only minor changes in their structure with variation of the parameter m . Stable clusters are generally isolated and compact. This is contrasted by weak clusters that lose their internal structure or disappear if m was increased.” This suggests a possible approach to gauge cluster stability. Hopefully the main clusters of interest with respect to the key results will prove to have good stability.

AR: We followed this approach and varied the FCM parameter ‘ m ’ (from the initial calculated $m=1.43$ to $m=1.53$ and $m=1.33$) and none of the clusters of interest in this study lost their internal structure nor disappeared, proving a good stability. But, please, see more details below.

Additionally, if I understand correctly each gene is assigned to each of the clusters with a score between 0 and 1 that sum to 1 (I believe that the colours in Figure S2 are designed to show these scores with high-scoring pink to low-scoring green and yellow genes). Cluster membership is then defined simply by the highest score (this assignment is not actually detailed in the supplement as far as I could tell, so this is my assumption). Thus, by eye in Figure S2 we can get a sense of which clusters have mostly strong membership (mostly pink) and which are more diffuse (more purple, blue, and green). Median score might therefore be a good numerical proxy for gauging the relative robustness of each cluster, e.g. it would clearly show that for clusters 12 and 18 presented in Figure 2b, cluster 12 scores are lower than cluster 18 scores. Note that when it comes to then test these gene lists for each cluster for Gene Ontology (GO) term enrichments these scores are ignored (see point [2] below for suggestions with respect to this).

AR: The Reviewer is right. Each gene is assigned to each of the clusters based on the score between 0 and 1 obtained through the Mfuzz function 'Membership', that is represented in a colour code manner, with high values and therefore, high correlation between the gene and the cluster, in magenta and low values (low correlation) represented with blue and green colours (we have now changed the grey scale clusters in the main Figures to colour code ones for a better visualization). In this regard, our Mfuzz analyses in general, and in particular the clusters in which our study is focused showed high robustness. In this way, the number of core genes (defined as those with membership 0.7 or higher and now represented in Supplementary Figure 2) in each of these clusters of interest (21, 18 and 30, followed by clusters 9, 10 and 12) was very high in all cases (> 80 core members). Only cluster 12 had a relative lower number of core genes, and even in this case, still 6 OBP genes out of 22 were among its core genes. We provided in Supplementary Table 8 and Supplementary Figure 2 the numbers of core genes per cluster as a proxy of the robustness of each of them.

Furthermore, as mentioned before, to assess the stability and robustness of the clusters and how this could affect our conclusions, we re-ran the analysis varying the FCM parameter 'm'. Since the estimated 'm' value (using 'mestimate' function) was 1.431382, we used as new values $m=1.33$ and $m=1.53$. In both cases, all the equivalent clusters to cl 21, cl 18 and cl 30 ('embryonic', 'nymphal' and 'adult', respectively) maintained their internal structure and further analyses showed a high level of overlap between the genes included in those clusters under the three different 'm' values (see new Supplementary Figure 7). Moreover, these three clusters maintain most of the top enriched GO terms when varying m, as it is shown in the tables (Fisher tests) in new Supplementary Figure 7.

The developers of WGCNA themselves stress the need to assess whether modules are preserved and reproducible (Is my network module preserved and reproducible? Langfelder et al. 2011 <https://doi.org/10.1371/journal.pcbi.1001057>). The focus of these discussions was actually on biological reproducibility between species rather than technical, but the approaches should still apply. They should also be useful for the mayfly-fly and mayfly-centipede comparisons as an alternative to the applied cross-tabulation approach (hypergeometric tests on overlaps between homologous groups for each pair of modules) for measuring module preservation without requiring module detection in the other species. Calculation of module preservation statistics is already part of WGCNA (modulePreservation function), so applying this type of analysis could provide additional evidence of cross-species module preservation.

AR: We thank the Reviewer for this suggestion since the new analyses have added further support to the cross-species module conservation.

We have now performed module preservation analyses following the tutorial at "<https://horvath.genetics.ucla.edu/html/CoexpressionNetwork/ModulePreservation/Tutorials/HumanChimp.pdf>". We think this analysis is appropriate for our datasets, as in this case the comparison is between datasets from different species as it happens with mayfly and fly, - although more closely related than our study. We obtained high scores (low p-values) of module preservation in modules such as greenyellow (Brain), cyan (Protein synthesis), turquoise (Ovaries), blue (Muscle), magenta (Male head-Phototransduction), tan (E10-Neurogenesis), yellow (Wing), paleturquoise (Gut), in a similar way as with our hypergeometric tests, validating the conservation of our obtained WGCNA modules. This validation of our previous results is even more remarkable if we take into account that Langfelder et al. developed the module preservation function by applying it between very closely related species (such as human and chimp) or even between different datasets

and/or individuals of the same species (in mice), where the vast majority of genes included in the WGCNA (or even all genes in the mice example) will have the required 1 to 1 equivalence to run the module preservation function. In our case, we are comparing very distantly related species, such as *Drosophila*, *Cloeon* and *Strigamia* whose lineages have evolved independently at least since the Carboniferous in the case of the two insects and the Cambrian in the case of *Strigamia*. Duplicated genes not showing a 1 to 1 relationship are of course much more prevalent here than in the human-chimp comparison. Thus, it is noteworthy that even taking into account only single copy orthologs, the module preservation function was able to recover the conservation relationships that we have observed before using the hypergeometric tests (in which all the orthologous genes are taken into account). This is the reason why we chose to use only the hypergeometric tests in the previous version, since we think that in WGCNA comparisons at deep evolutionary distances (which were not the aim of Langfelder et al 2011 and are not frequently found in the literature) is important to have as much information as possible and being able to compare all orthologous genes, even if duplicated (this is even more crucial in cases with whole genome duplications, as in a previous study comparing amphioxus and vertebrates in which some of the authors participated).

We have included now a sentence in the main text regarding these new results “Importantly, we also found this deep conservation using an alternative metric for module preservation and in-group proportion statistics³³⁻³⁵. This further supports our results (Supplementary Figure 6b).”. We have also new panels showing in the Supplementary Figure 6 the heatmaps with these results and the corresponding methodology in Supplementary information.

One possibility to confirm the robustness of the WGCNA modules themselves could be to apply an iterative approach e.g. iterativeWGCNA (<https://www.biorxiv.org/content/10.1101/234062v1>).

AR: We tried to apply the iterativeWGCNA described in the preprint suggested by the Reviewer, but despite the investment and considerable efforts in our hands we were unable to implement and run the approach described in that prepublication. Therefore, we decided to apply an alternative strategy to confirm the robustness of our WGCNA modules. We used five different replications of the analysis in *C. dipterum* and we observed no substantial changes in the module assignments. We performed five new analyses by modifying the default parameters BETA 6, VAR 1 to Run2: BETA 12, VAR 1, Run3: BETA 18, VAR 1, Run4: BETA 6, VAR 5 and Run5: BETA 6, VAR 20 and the vast majority of genes still belonged to the same module. We included the results of these replications as an additional sheet in Supplementary Table 12.

[2] gene ontology assignment and enrichment

The qualitative descriptions of the functional roles of various clusters and modules come from the tissues and/or life-stages where they are expressed combined with hints from Gene Ontology terms and PFAM domain annotations. These are clearly key to many of the results presented, but I could not find details of how GO terms were assigned, and PFAM seems to cover only a quarter of all *Cdipt* genes. For example, the 33-page WGCNA summary PDF simply presents each module with expression across the samples, topGO results, and the ‘name’ assigned, for *Cdipt*, *Dmela*, and *Smari*. I could not find any details about where the GO-term annotations comes from. I presume *Dmela* would be sourced from the

corresponding gene set version at FlyBase, but how were Cdipt and Smari genes assigned GO terms?

AR: We have included now in Supplementary information (1.5 Gene annotation) details of how the GO terms were assigned by using the *D. melanogaster* gene to GO database from Ensembl BIOMART: "Taking the orthology (from orthofinder) from *C. dipterum* or *Strigamia maritima* to *Drosophila melanogaster* (taking the *D. melanogaster* gene to GO database from Ensembl BIOMART, and pulling across all the GO terms for each gene), we generated a topGO gene to GO key, by copying across all GO terms represented in each orthogroup from *D. melanogaster* to *C. dipterum* or *S. maritima*, respectively. Functionally annotated mayfly (or *S. maritima*) proteome was utilised for subsequent analyses (e. g., Enrichment analysis for Gene Ontology using topGO package (v 2.36.0)¹⁹)"

Regarding PFAM annotations, we rephrased the supplementary information section to clarify that 11,030 genes out of 16,364 (67%) contained at least one of the 4,308 PFAM domain models: "These models contain 4,308 non-redundant PFAM *domain* models in 11,030 genes as assigned using the PfamScan tool." (see also the additional comments on the PFAM domains below).

Additionally, there are no details presented about the actual tests run using topGO (which test statistics and algorithms – classic, elim, weight, parentChild, fisher, ks, etc.), and how the background was controlled – because only expressed genes can be assigned to a module it means that the background can only contain expressed genes not all genes.

AR: We have now included all these details and significantly expanded the section in Supplementary information regarding the topGO analysis (two-sided Fisher's exact tests) and the background that was used for each of the analyses: "...Taking the functional annotation of *C. dipterum*, (see methods 1.5 annotation) we generated a topGO gene to GO key. We performed an Enrichment Analysis for Gene Ontology using topGO package (v 2.36.0)¹⁹ and used all genes considered in the Mfuzz experiment as a background. Uncorrected *P* values shown in plots corresponded to two-sided Fisher's exact tests as provided by topGO¹⁹."

Methods for GO-term enrichments of Mfuzz clusters are described in the supplement with one sentence: "Taking the *Drosophila melanogaster* orthologs, we performed Gene Ontology (GO) enrichment analysis for each of the clusters using DAVID". Given how the GO-term enrichments form such an important part of the results, helping to define the main functions of various clusters, this description is woefully inadequate.

AR: We re-ran our GO enrichment analysis using topGO to maintain the same type of analyses in both Mfuzz and WGCNA clustering and include a thorough description in the Supplementary Information. The results obtained with topGO show significant values (two-sided Fisher's exact test) and the GO term categories still reflected the main biological and molecular processes occurring for each of the clusters.

(1) While using *Drosophila* orthologues to get at function is fine I would also like to see evidence that similar terms can be identified when using Cdipt gene annotations directly because there will be many cases where orthology is not one-to-one.

AR: We are not sure what the Reviewer means with direct *C. dipterum* annotations, since before our work, there were no functional annotations (associated to Gene Ontology terms)

for this species and therefore, it was necessary to use functional annotations from another species to obtain GO term enrichments; we used *Drosophila*, the closest well-annotated organism.

We generated the GO key by getting and assigning all the GO terms (from Ensembl BIOMART GO database) from all *Drosophila* genes represented in each orthologous family. Therefore, the analysis is applied to the orthologous families and not to individual loci, which means that in cases where there are several *Drosophila* orthologs, the corresponding *C. dipterum* gene(s) included in that orthologous family will get all the GO terms. Nevertheless, the results we obtained in all our analyses were biologically meaningful, validating this approach.

(2) There is no mention of the background gene set, was this correctly controlled? The foreground, for each cluster, is a list of *Drosophila* genes with orthologues in *Cdipt* that were expressed in *Cdipt*. The background therefore should not be all *Drosophila* genes provided by DAVID, but rather all *Drosophila* genes with orthologues in *Cdipt* that were expressed in *Cdipt*.

AR: We used as background all genes expressed (used in WGCNA and Mfuzz respectively), now mentioned explicitly in the text. We modified the Mfuzz method section as follows: " We performed an Enrichment Analysis for Gene Ontology using topGO package (v 2.36.0) ¹⁸ and used all genes considered in the Mfuzz experiment as background. Uncorrected *P* values shown in plots corresponded to two-sided Fisher's exact tests as provided by topGO ¹⁸ ". As commented before, to keep consistency across the different analyses, we have not used DAVID in the current version of the manuscript.

For confirming GO-term enrichments for Mfuzz clusters: currently I believe all cluster members are treated as the foreground for enrichment testing, I would suggest (1) eliminate low-scoring genes from the list before testing (but this annoyingly involves selecting a cut-off, and applying the same filter to the background gene list); alternatively (2) using a score-based enrichment test such as the Kolmogorov-Smirnov test implemented in the topGO R package would remove the need to select a cut-off (also known as gene set enrichment analysis, GSEA). Note that the full background gene list should not be all *Cdipt* genes, but it should consist of all genes assigned to any of the clusters (this was not detailed in the methods so I cannot assess whether the background was correctly controlled or not in the current analyses). If the GO terms shown in Figure 1d are still significant then these are robust results.

AR: For the GO-term enrichment for Mfuzz clusters we have used a score-based enrichment test, two-sided Fisher's exact test implemented in topGO R package, as suggested by the Reviewer (See methods and Supplementary information). In this case, the background was the total of functionally annotated *C. dipterum* genes that were assigned to the Mfuzz clusters and still, the GO terms obtained (regulation of gene expression, DNA binding, cuticle development, odorant binding, or detection of light stimulus, among others) were significant, further supporting the robustness of the results.

Minor points.

Please note, not all minor points are listed here. I provide a scanned copy of my comments on the main text, and I also provide a copy of the supplement with tracked-changes and further comments for consideration.

AR: We carefully checked the scanned copy of the manuscript and the supplementary information and modified them accordingly. Among all these detailed comments, in the Supplementary Information there were a few points from the Reviewer that suggested deleting some sections containing extra information that was not explicitly referred to in the main text. Although these sections are not so central and are less connected to the main focus of our study, we believe they contain valuable information that could be of interest for some readers, therefore we would prefer to keep them.

Abstract: while I realise it is difficult to get all major results to fit into the abstract, I feel that the gill-wing result should appear in the abstract.

AR: We have included now a sentence in the abstract referring to this result: "Furthermore, we observe transcriptomic similarities between gills and wings, suggesting a common genetic program for these two organs."

Please make sure all acronyms are defined on their first usage, this is very important for readability, e.g. Myr; CEGMA; BUSCO; HRP; WGCNA; ONT; HMM; DAPI; etc.

AR: We have now included all the definitions on their first usage.

Please make sure all panels of all figures are correctly referenced from the main text, e.g. Figure 2b and 4b do not appear to be referenced in the text. Also, the use of primes, e.g. a and a', to distinguish some panels is hard to see – please use distinct letters instead.

AR: Figures 2b and 4b are now cited in the text: "...and 18 are nymph-specific (Figure 2b)." and "...including developing wings, for *C. dipterum* and *D. melanogaster* (Figure 4b and see..." We have also modified the panel names trying to reduce the use of primes (see Figure 1, Fig 3 and Fig 4)

Introduction: while I appreciate the brevity as it stands now, it would seem appropriate to mention current thinking on the origin of wings somewhere in the introduction.

AR: We have included the following sentence and corresponding references in the introduction: "...the different hypotheses accounting for the origin of wings, which suggested that wings are either homologous to tergal structures (dorsal body wall), or pleural structures (including gills) or a fusion of the two."

Figure 1: the title does not reflect the content – what does the figure show about the genome?

AR: We changed the title to: "The mayfly *C. dipterum* and its transcriptomes throughout its life cycle"

L123-127: genome size estimates should probably also be mentioned in the main text, not just the supplement. In the supplement clearer details on how k-mers were chosen need to be provided to make sense of the analyses performed.

AR: We have now included this information. The different estimates of the genome size were conducted using k=17 or k=31. Now this is described in the supplementary material and summarized in a new sheet in Supplementary Table 1. Although these are default values, they are bracketing around k=21, a value that is often recommended, because it is long enough to span repeats (provide more unique k-mers) and short enough to include fewer sequencing errors. Pushing the k-mer lengths above 31 will increase the amount of unique k-mers at the cost of having lower k-mer coverage and requiring higher memory for computation. We think these four estimates should be enough to provide a good idea of the actual genome-size of the species. In fact, our final genome assembly falls in the range of genome size estimates extracted from the paired-end data.

L128: was assembled in 1395 scaffolds, with N50 of 0.461 Mb. But please check because in SOM it says CONTIG N50 = 461,411 bp and SCAFFOLD N50 = 434,950 bp. This is strange – how can scaffold N50 be shorter than contig N50, please double check.

AR: Thanks for noticing the mistake, the numbers were switched. It is corrected now.

L130: is the CEGMA assessment of the gene set or of the genome assembly? Same for BUSCO, the text suggests the gene set was assessed, but the supplement describes the assembly assessment, what were the results for the genome assembly and what were the results for the gene annotation – please clarify exactly what is being presented. Also, please present full BUSCO results, i.e. include duplicated and fragmented percentages as this is needed to give a full picture. E.g. if % duplicated is alarmingly high then the assembly could have failed to collapse haplotypes, which would then raise questions about the reported result of the large number of OBPs that were identified. For example, the opsins section indicates that “opsin sequences were carefully inspected and curated to detect gene annotation errors such ... and identical sequences derived from haplotypes of the same gene” which suggests that there were some ‘haplotypes’ – is this an issue or not, please address.

AR: Both CEGMA and BUSCO were used to evaluate gene completeness of the assembly. Is true that the sentence in the main text was confusing, we have now addressed it. Also, we certainly omitted some of the figures in order to simplify the main text, but now the percentage of each class in the orthologous searches are reported for CEGMA and BUSCO in the supplementary. The percent of complete duplicated BUSCO is now mentioned in the main text. In the supplementary now says: “BUSCO v3 (Simão, Waterhouse et al. 2015) using a more comprehensive database (insecta_odb9) that includes up to 1,658 BUSCOs, we found 96.9% Complete BUSCOs (94.1% in single-copy and 2.8% Duplicated), 1.3% Fragmented and only 1.8% missing (29 genes). In summary, 98.7% of the insecta_odb9 BUSCOs are present in our clodip2 assembly.” Regarding to the Reviewer concerns about possible duplication issues in the assembly, we consider the number of duplicated BUSCOs found to be moderate rather than alarmingly high. However, as *a posteriori* sanity-check we quickly have run *pseudohaploid* (<https://github.com/schatzlab/pseudohaploid>) on clodip2. This is an alignment-based tools that allows identifying and discarding small contigs contained in larger ones. In this manner, we could obtain a “more haploid reference”. However the BUSCO results using insecta_odb9 are very similar; C:96.9%[S:94.3%,

D:2.7%],F:1.3%,M:1.8%,n:1658. The difference is mainly due to a change in category of 2 BUSCOs that appear now to be as single-copy after the modest *haploidization* of the assembly (see new supplementary version,):

1607 Complete BUSCOs (C)
1563 Complete and single-copy BUSCOs (S)
44 Complete and duplicated BUSCOs (D)
22 Fragmented BUSCOs (F)
29 Missing BUSCOs (M)
1658 Total BUSCO groups searched

The pseudohaploid reference has 1,563 Complete Single BUSCOs instead of 1,561. This is actually a minor improvement and is hard to tell if the nature of this duplications is artefactual or a real biological fact. But in any case, keeping this result in mind, the resolution of haplotypes in our assembly appears to be high and we would not expect to be introducing any major biases in the analyses detailed in our manuscript.

Regarding the haplotypes mentioned in the opsins section, all identified cases corresponded exclusively to *E. danica* sequences, we have not found any single case of putative haplotypes among all the *C. dipterum* genes that we have manually checked and curated in this study (i.e. all opsins and chemosensory genes). To clarify this, we have corrected the sentence mentioned by the Reviewer:

“All mayfly LWS, UV and Blue opsin sequences were carefully inspected and curated to detect gene annotation errors such as fusions and fissions of gene models, and missing or spurious exons. Furthermore, in the case of *E. danica*, we detected and corrected several cases of identical sequences derived from haplotypes of the same gene”.

The absence of uncollapsed haplotypes in our *C. dipterum* genome assembly is also to be expected given that the genome was sequenced from individuals cultured in the lab after 7 generations of inbreeding, in contrast to *E. danica*, which was sequenced from an individual captured in the wild.

L133: In Table S4 please give actual versions of annotation sets used, not just source DB, e.g. *Caenorhabditis_elegans* Ensembl; *Heliconius_melpomene* Ensembl; *Ixodes_scapularis* Uniprot; etc. are not useful for replication.

AR: Table S4 has been modified to include the names/codes of the release versions of annotation sets used. We have also added the corresponding genome publications.

L133-135: it appears strange to mention gene gain and loss here but not present it as a result per se “allowing the study of gene gains and losses at the origin of winged insects” is all we get here, and then all of a sudden much later in lines 309-311 there is a return to the gain/loss results. If there is a result to present for Figure 1a then present it properly.

AR: We have removed the sentence “allowing the study of gene gains and losses at the origin of winged insects” in lines 133-135 to leave the results referring at gain and loss of genes together in lines 309-311: “Moreover, we studied gene gains and losses at the origin of winged insects (see methods, Figure 1a). We did not observe any enrichment of new genes expressed in the wings in...”

Methods to infer gene gains and losses are not clearly described. The supplement simply points to a reference that seems to be about opsins. The section on gains/losses ends with “The *C. dipterum* Pfam and GO annotations were used to inspect the functions of these genes.” Where did the GO annotations come from?

AR: The Reviewer is right, there was a mistake with the literature cited, thanks for spotting this. We modified it (Bowles et al. 2020, Guijarro-Clarke et al. 2020 and Paps and Holland, 2018). Also, we added information about the method used: "...based on the taxonomic occupancy of the gene clusters. Briefly, novel genes are gene families present in at least one species of each of the two main lineages of clade (e.g., for Hexapoda, present in at least one paleopteran and one pterygotan) and absent outside the clade of interest. Core novel genes are genes present in every single lineage of a clade or absent only once, and are absent in taxa outside the clade of interest. The *C. dipterum* Pfam and GO annotations were used to inspect the functions of these genes in Paleoptera (see functional annotation section 1.5); for the rest of the nodes, the functions of the genes were inferred by submitting *D. melanogaster* IDs to PantherGO ³²"

L137-142: for transparency please clearly indicate in the main text which samples had replicates and which did not, which were PE and which were SE. This is given in the supplement but it currently feels hidden beyond merely for the sake of brevity.

AR: We have now indicated in the main manuscript the number of replicates and the sequencing strategy (PE or SE) for each of the RNA-seq samples in the corresponding figure legend. Furthermore, we have also included detailed information in the supplementary table (Supplementary Table 6) and supplementary information. We also tried to add this information to the corresponding paragraph in the main text as the Reviewer suggested, but this created a lengthy “laundry list” paragraph full of brackets with PE/SE info that compromised the readability of the text, so instead of this we decided to point to the readers that they can easily check this information in the figure “...see Figure 1 legend and Supplementary info for details on the number of replicates and sequencing”). However, if the Reviewer still thinks that there should be a paragraph in the main text listing this, we will add it.

There were a couple of places where I wondered whether the lack of a replicate for E6 would have an impact on the type of analysis that could be performed – please indicate clearly where E6 is included or excluded because of having only one replicate and openly describe this (and the single replicates for nymphal heads) in the supplement.

AR: E6 is the only case for which we only had one replicate, extracting RNA from these early embryos was technically very challenging and finally only one of the samples of stage 6 embryos was obtained. Despite the lack of a replicate, we decided to use it because we thought that the E6 stage (segmented embryo) could be useful for having a more complete developmental time course. However, before taking this decision we made sure that, as the Reviewer pointed out, this did not have an impact in our analyses. First, this study does not focus in any aspect specifically related to the biology of the E6 embryonic stage, so our results and conclusions were not likely to be affected. Second, and more importantly, the results obtained (for example Mfuzz clustering) do not show any discrepancy with respect of what can be expected for this stage that could indicate that this single replicate could be an outlier. In other words, a priori, we expected that E6 sample should behave similarly to E4 and E10 replicates, as E6 stage (segmented embryo) is a bit more advanced stage of

embryogenesis than E4 (germ disc embryo) and a bit more distant to the following stage, E10, so we should see smooth transitions and shared features. Indeed, this is the case for Mfuzz profiles (where gene expression progresses as expected, without bumps or valleys at this particular stage), WGCNA (in all the modules obtained the 2 E4 replicates and E6 sample behave similarly) and the transcriptomes clustering (the 3 embryonic stages clustered together). E6 was only excluded from the wing-gill comparison because developmental expression was already being taken into account with the inclusion of E4 and E10 and we considered that it was important for this analysis to be sure that our expression cut-offs were consistent across 2 replicates.

Regarding the nymphal heads, we detail now in the figure legend of the main text and in supplementary Table 6 that indeed, these 10 samples contained replicates: 4 replicates for early, 2 replicates for mid and 4 for late nymphal heads.

Also, why was Female Head replicate 2 excluded from Figure 4e?

AR: When performing Principal Component Analysis (PCA) and clustering of the samples, one of the Female Head replicates showed an unusual pattern: instead of clustering together with Male heads or Female Brains, it was close to some embryonic samples. This could be explained by the fact that *C. dipterum* is an ovoviviparous species and that most of the adult female body mass is dedicated to the reproductive function: almost the entire body cavity of females is full of developing embryos, so, during the extraction of the RNA, some embryonic RNA could have been also captured. As a result of this, we decided to remove this sample from those analyses in which the main focus was to compare gene expression across tissues as in the Figure 4e (now 4g, mentioned by the Reviewer) and in the module definition for WGCNA analysis. However, we kept it for the cases in which it was particularly important to check gene expression in heads (differential gene expression between male and female heads or expression of chemosensory genes). We have clarified this in the corresponding sections of the supplementary information

Particularly for readers interested in technical aspects, please add a brief mention in the main text and some commentary to the supplement on why, given 96x Illumina coverage and 36x Nanopore coverage (with N50 3.5 Kbp) and inbreeding (e.g. did this work?), the contiguity of the resulting hybrid assembly was not really outstanding: ~1400 scaffolds N50 ~460 Kbp. E.g. looking at scaffold-level assemblies at NCBI of similar sizes (150-210 Mbp) with similar number of scaffolds (1000-3000), which so far as I can tell did not have any long reads, all have higher N50s: *Fopius arisanus* 978,588; *Polistes dominula* 1,625,592; *Orussus abietinus* 612,083; *Drosophila obscura* 472,512; *Cephus cinctus* 622,163; *Galendromus occidentalis* 896,831; *Drosophila pseudoobscura* 1,459,550.

AR: Although the sequencing coverage used for both Illumina and nanopore would be optimal for hybrid assembly with MaSuRCA, the efficiency of the assembly (in terms of contiguity) is also strongly limited by the read length distribution of ONT reads and the genome architecture of the organism. The read lengths of the ONT reads are limited not just by the molecular weight of the DNA extraction but also by its integrity. Double-strand DNA molecules with nicks in one of their strands often will break while passing through the pores limiting the final read length. The fact that we needed to pool several individuals in order to obtain enough DNA to achieve 36x coverage might introduce breaks in the DNA as reflected in the short N50 3.5Kb observed for ONT. While we normally observe read length distributions of 7-12Kb, insects and other organisms with challenging extractions tend to exhibit lower read length N50s. By having lower read lengths, we ended up with shorter

mega-reads and this explains that final contiguity does not reach the mega-base scale. As another Reviewer pointed out we could improve our assembly contiguity in the future, pushing the assembly to chromosome-scale, by adding Hi-C data. However, the level of contiguity of our current assembly version did not limit the gene completeness and the analyses conducted here. In our opinion, 0.435Mb contig N50 provides enough genomic context for the annotated genes, especially in the case of this genome, which is relatively compact (180Mb).

Regarding the comparison with these other genome assemblies obtained without long-reads, we have to say that most of them have a much lower N50 as stated in GenBank (see new sheet in Supplementary Table 1). The only exception appears to be *Oabi2.0*. However, we apply a much stricter definition for contigs as they are depleted of any gaps. Our assembly indeed shows fewer gaps and they account for 18.4Kb instead 83.9Kb. After breaking the assembly into contigs at gaps of 1 base pair or longer*, the contig N50s are 124.8Kb and 434.95 Kb, respectively. Therefore, the continuity of the mayfly reference used here is still 3.5-fold higher than the one observed in *Oabi2.0*.

* *Our criteria might sound aggressive but in fact some assemblers place 1 N as unknown length gaps and with the given sequence-quality of Illumina is quite unlikely that all these Ns correspond to ambiguous bases.*

Annotation: from supplement "... Augustus gene prediction tool which yielded 16,364 evidence-based gene models ... These models contain 4,308 non-redundant PFAM models as assigned using the PfamScan tool" – that is just 26%, is this normal/expected? Would InterProScan give better domain assignment coverage than just PFAM?

AR: 4,308 PFAM models referred to the total number of pfam domains, not the number of genes with a pfam domain. Following the same procedure, *D. melanogaster* has 4,333 pfam domains. PFAM is usually a rather basic classification (hence its strength to assess diversity). For instance, all homeobox genes will have one Homeobox PFAM. Therefore, it is normal to obtain less PFAM than genes and we considered that InterProScan would not improve significantly these numbers. In the case of *C. dipterum*, 11,030 genes have a PFAM annotation, which is 67%. Comparatively, *Drosophila* has 10,792 genes with one or more PFAM domains.

Also, transcriptome mapping identified about double the number of putative gene loci, so why was the final prediction so much lower?

AR: The final prediction for the transcriptome cited in the text referred only to protein-coding genes, whereas the putative gene loci included non-coding loci as well (e.g., lncRNA). To clarify these data, we modified the following sentences in the Supplementary information: "...gene prediction tool which yielded 16,364 evidence-based *protein-coding* gene models." And "These models contain 4,308 non-redundant PFAM *domain* models in 11,030 genes as assigned using the PfamScan tool."

Mfuzz vs. WGCNA: as some readers will not be too familiar with such gene expression clustering methods, for clarity please briefly indicate in simple language why Mfuzz was used for time-course gene expression clustering and WGCNA was used for delineating modules of co-regulated genes across tissues. This should be then elaborated on in the corresponding supplementary text sections, which are currently also severely lacking in detail. For example, a key feature of Mfuzz is that genes can belong to more than one cluster – not all readers will

be aware of this and it is an important fact when trying to interpret the figures and main results. I'm guessing here that the best-scoring cluster was used to define membership, but this is not detailed as far as I could see.

AR: We have completed the information regarding the two different clustering tests (Mfuzz) and (WGCNA) in the supplementary information. We included here details on how genes were assigned to Mfuzz and WGCNA clusters based on high scores of membership values. Also, we modified the grey-scale Mfuzz plots by colour-based plots to help visualising the robustness of each cluster.

Mfuzz methods: the level of detail given is totally inadequate. What fuzzifier 'm' value was used? And how was it selected? What data exactly were used? Counts? Were they log-transformed? E.g. Mfuzz website says "Since it is common to log-transform the FPKM before clustering (although it is not absolutely required) ..." Was Mfuzz standardisation also carried out? And what about low/no expression genes? E.g. were genes with low/no values in most samples first filtered out? How were 'missing values' dealt with? How exactly was the optimum number of clusters decided? Then to finish, what do the results actually look like? E.g. How many genes, and what level of overlap was there amongst the resulting clusters? Table S8 seems to indicate that Cdip genes each belong to only a single cluster – how was this decided, and what cut-off was applied for a gene to belong to a cluster or not? What proportion of cluster members are considered core members, e.g. applying alpha core cut-off of 0.7? How can cluster robustness be demonstrated?

AR: We have expanded considerably the amount of information regarding the methodology used to generate the soft clusters using Mfuzz: data used (normalised counts), pre-processing of the data (filtering and standardisation), Dmin results to decide optimum cluster number, how the FCM 'm' was calculated (mestimate function), etc. We believe that now the methodology is clearly explained in Supplementary Information. Furthermore, we included the number of genes and core genes (with an alpha core cut-off of 0.7) in Supplementary Figure 2 and we generated a new supplementary figure (Supplementary Figure 7) showing the robustness of the clusters when we modified the parameter 'm'.

L151: here and elsewhere, please be very careful when using the term 'specifically' or 'specific' when referring to expression patterns. Normally 'specific' expression of certain genes suggests that these genes are not at all expressed anywhere else. Please rephrase here and elsewhere to be clear about exactly what is meant when describing these expression patterns.

AR: We have rephrased the following sentence: "focused on clusters containing genes whose expression peaks at each of these phases " instead of the previous version "...clusters specifically in each of..."

L173: proteins do not expand, families expand, please rephrase. Also, L222 and elsewhere.

AR: We are sorry for this, OBP genes are a tricky case because the gene family name is 'Odorant Binding Protein', so it is quite common to make mistakes and/or to misinterpret it. We have modified the title to clarify this, taking also into account the suggestions of Reviewer 1: "Expansion and role of Odorant Binding Protein genes during the aquatic phase", and also changed "opsins" by "opsin genes".

For gene family and orthology assignments, if some of the species do not yet have a 'genome publication' to reference then it would be appropriate to acknowledge people or consortia who granted permission to use such pre-publication data in the main acknowledgements section.

AR: All the species included in our analyses had a previous genome publication, the only exceptions were *E. danica* and *L. fulva* but in these cases the coordinator of their respective genome projects, Bernhard Misof, is one of the authors of this manuscript. Furthermore, while our manuscript was under revision, the genomes of these two species have been published. However, we thank the Reviewer for stressing this important point since in the previous version many of these citations were missing. We have now made sure that the genome publications of all these 28 species are properly referenced as well as included in Supplementary Table 4 together with the name of the corresponding release versions. Furthermore, Dr Andrea Luchetti, who provided the annotated proteome of *L. arcticus*, which was not available in the original publication, has now been included in the acknowledgements section.

L181-189: please be honest about the numbers of complete gene models identified, there are several discrepancies between what is reported here in the main text and what is presented in the supplement. Also, the gene annotation pipeline produced alternative transcripts, yet for these gene families of biological interest no mention is made of whether the protein repertoire is actually larger than the gene repertoire as would be the case through alternative splicing.

AR: We apologize for these discrepancies, they were obviously unintended. The number of gene models is now corrected. Regarding alternative splicing, as the main interest was to characterize the gene repertoire for each chemosensory-related gene family –and that we considered the different isoforms as hypothetical/putative given that they were generated by automatic annotation-, we clustered them in a final set to obtain unique gene models. Furthermore, most CS complements characterized in other insects have also been done at the gene locus level, so we also focused in the gene repertoire rather than the isoforms because it was more readily comparable with previous literature. The list of putative proteins (isoforms) predicted for each family is:

OBP 191 genes - 310 putative proteins

CSP 16 - 21

GR/OR* 64/50 – 132

IR/iGluR 34 - 114

SNMP 12 - 38

(*GR and OR gene families are grouped together as the structural domain of both families is similar and it is only possible to classify them phylogenetically in some cases)

However, we did not include these numbers in the manuscript since they are approximate numbers without additional experimental validation.

L191: the use of the word 'previous' here and elsewhere, suggests results from a previous published study please rephrase to be clear about exactly what it meant. Also, L304.

AR: We eliminated the word 'previous' and rephrased both sentences: "Our GO term enrichment analyses" and "..., on the results obtained through the WGCNA modules..."

L199: nymph-specific, again, careful with the use of 'specific', especially here because clearly clusters 18 and 12 show first increase in expression already in E14, i.e. technically before the nymph stage.

AR: We rephrase now as follows: "... and genes in clusters 9, 12 and 18 were most highly expressed during nymphal stages (Figure 2b)."

L206-207: 34% - please check, this seemed confusing with respect to the numbers presented in the sentence. Also, I could not find the methods for building the heatmap shown in 2c.

AR: We have modified the sentence now to clarify where this number comes from, as the total number of chemosensory genes was omitted before: "...however, an additional major chemosensory tissue, the gills, where 34% of the 276 CS genes were expressed, (5/16 CSP, 8/34 IR and 82/167 OBP genes, Figure...". Moreover, the method to build the heatmap is now described in Supplementary information (4.9 Gene clustering and heatmap plots).

L235-240: 'expansion' why not just state that there are 4 copies?

AR: Changed

L246: "whose protein sequence is highly modified" how does either 3b or 3c show how the protein sequence is modified?

AR: Panels 3b and 3c (now renamed as 3c and 3d) referred to the differential gene expression of *UV-Ops4* and *Blue-Ops2* and the snapshot of the genome browser showing the two copies of blue opsin genes and their differences in expression throughout different RNA-seq samples, and not to the fact that the second Blue Opsin sequence is divergent. We moved the brackets just after *blue-Ops2* and left the rest of the sentence (which now explains better why we consider it a highly modified protein) behind them to make clear that the figure shows these results: "... *UV-Ops4* followed by *blue-Ops2* (Figure 3c, d, Supplementary Table 11), a divergent blue-Ops duplicate comprising only the N-terminus part of the protein containing the retinal binding domain. "

WGCNA methods: The level of detail given is totally inadequate.

AR: We expanded the information regarding the WGCNA methodology in Supplementary information: "...To construct the weighted gene network (co-expression similarity), we used softpower settings of 6 for *C. dipterum*, *D. melanogaster* and *S. maritima* datasets. Calculated 'adjacency' was utilised to construct the Topological Overlap Matrices (TOM) and their corresponding dissimilarities. After performing the hierarchical clustering with the 'hclust' function, we set a minimum module size of 30. Finally, we merged modules with similar expression profiles using as height cut 0.25 to obtain 22, 17 and 21 modules for *C. dipterum*, *D. melanogaster* and *S. maritima*, respectively. These modules contained genes that were assigned to them based in their membership values across the RNA-samples used in the study..."

WGCNA recommends log-transforming RPKMs – was this done?

AR: Although it is recommended, it depends on the pre-processing and previous normalization of the datasets. As our RNA-seq samples were not too heavy-tailed and random errors were generally symmetric, we used corrected RPKMs without log-transformation.

WGCNA website “We do not recommend attempting WGCNA on a data set consisting of fewer than 15 samples.” 14 for *Strigamia* is only just below their recommendation, so it might be alright, but this should surely be mentioned.

AR: As the Reviewer highlights, 14 is just below the recommendation. Despite of the limitation on the number of samples (due to sample availability), we considered worthy to perform the analysis in this species since *S. maritima* is an outgroup (non-insect arthropod) to the other two species investigated in this work. Indeed, the results we obtained were biological meaningful despite the limited number of samples. We now explicitly mention this in the supplementary information: “Note that the recommended number of samples to perform WGCNA analysis is 15 and only 14 samples were available for *S. maritima*. However, when analysing the modules obtained for this centipede species, we were able to assign different ‘categories’ based on either high gene expression in a main tissue and/or GO term enrichment to different modules representing meaningful biological processes or functions.”

The sample information in Table S11 is confusing: Sample counts do not correspond to sample lists. *Cdip* samples: 26, but 27 in list (one female head excluded – why, because no replicate, but E6 is in the list of samples?) *Dmel* samples: 19, but 16 in the list, e.g. *Adult_Fem_Heads* and *Adult_Male_Heads* and *Embr_12_14h* and others are not in the list *Smar* samples: 14, but 7 in the list.

AR: There was missing information in the Supplementary lists for the WGCNA information. Now it is corrected and information about WGCNA modules is added in Supplementary table 12 (before supplementary Table 11)

How were all these samples chosen and what effect does the choice of sample have on the resulting modules? I.e. if you change the input data do you still get to see the same major modules appearing or is this highly sensitive to the data?

AR: We aimed to have as many samples as we were able to obtain and sequence to get a broad variety of different tissues and developmental/embryonic stages. In the case of *C. dipterum*, we used all the samples available, except for one of the replicates of the female head (see above) and the nymphal heads because we wanted to avoid having an unbalanced sample selection where many samples corresponded to the same tissue (the inclusion of these samples would have meant 11 heads out of 34 samples) that could bias the whole analysis. For the other two species, we selected available samples that could be equivalent to the mayfly ones in terms of tissue or embryonic stage. We do not think that our choice of samples had major impact on the resulting modules and probably just when reducing significantly the number of samples under the recommended number it would have

promoted an increase in the noise of the analysis. Moreover, when we changed the WGCNA settings, still each module only related to one (see comment above).

Filtering: “We performed the analyses using as datasets genes that were present in a least two species in our family reconstructions and showed variance across samples” – if I understand this correctly it means that any genes in *Dmela*, *Cdipt*, and *Smari*, that do not have orthologues in at least one of the other species would be filtered out, correct? Please explain the reasoning, and what ‘family reconstructions’ are – OrthoFinder with 14 species or 28 species?

AR: We thank the Reviewer for spotting this, since this is actually a mistake in the methods description (this orthology filtering was applied in previous works but not in this study). Thus, we did not apply any restrictions in terms of orthology, in each of the three species we used all the genes that showed variance > 1 across samples, which in all cases corresponded to the vast majority of genes (13,720 in *C. dipterum*, 12,463 in *D. melanogaster* and 13,001 in *S. maritima*). The sentence now reads as follows: “We performed the analyses using as datasets all the genes that showed variance across samples (coef. var ≥ 1).” We apologise for this mistake.

L272: the methods for these tests are not described in sufficient detail to assess or reproduce.

AR: We have now detailed in the Supplementary information how we performed the hypergeometric tests in the pairwise comparisons between the three species:” To evaluate the significance, we performed hypergeometric tests. We translated the list of genes contained in each module into a list of ‘orthologous gene families’ (obtained from our orthologous families from Orthofinder2). In this way, two genes that belonged to the same family were both translated to their family’s identifier. This allowed us to compare modules across species.

After this ‘translation’, we kept a set of orthologous gene families per module and then considered a hypergeometric test where the number of orthologous gene families (gf) in one module was the sample size (SS), the number of gf of the other module was the ‘success in population’ (SIP), their common gf was the ‘success in sample’ (SIS) and the total available gf (including those found in neither of the two modules tested) was considered the population (POP). In other words, we computed a probability of finding SIS, or more, common families in our sample of SS families (the one module), that also belonged to the other module which contained SIP families out of POP total available families.

```
a = set([module 'a' gene-family-ids])
b = set([module 'b' gene-family-ids])
```

```
gene_SS = len(a) # Sample Size
gene_SIP = len(b) # Success in Population
gene_SIS = len(a.intersection(b)) # success in sample
gene_POP = total number of gene family unique ids
hsv = scipy.stats.hypergeom.sf(gene_SIS-1, gene_POP, gene_SS, gene_SIP)
value_in_heatmap = -log(hsv, 10)”
```

L277: clustering methods that produce the dendrograms and hence determine the orders of tissues and life-stages presented are not explained.

AR: We added the methodology to produce the different dendrograms and tissue/life-stages clustering in the supplementary information ('3.3 Weighted Gene Correlation Network Analysis (WGCNA)').

Fig 4a Malpighian tubule is incorrectly shown as tube.

AR: Corrected

There is more here to discuss: testis-testis is weak, but present; ovaries-ovaries are nice and strong; head/vision seems good; but how to explain ovaries-wing discs and testis fat body?

AR: We have to take into account that, despite the module conservation we have found, these two insects might have important anatomical and physiological differences, many of which may have not been studied in detail so far. Although the testes-fat body connection may look strange, it has been shown that some species of insects (as seen in Ren et al. *Insect Biochemistry and Molecular Biology*, 2019, for example) have fat body-like tissues intimately associated to testis follicles that are essential for spermatogenesis. Therefore, a similar association/function could be present in the testis of mayflies. However, to confirm that this is the case for *C. dipterum* as well, it should be investigated in much more detail with anatomical and molecular studies that, in our opinion, are out of the scope of this work. In the case of the ovaries-wing discs connection, we have to consider that the wing disc module contains a much larger number of genes (1919) than the wing pad module of mayfly (699), therefore, in addition to the genes responsible for the commonalities that promoted the correlation between these two modules, the wing disc module has genes that, considering the GO term enrichment analysis, are involved in RNA processing in one of the most proliferative tissues of the fly larvae, the imaginal discs, which coincides with the type of genes expected (and found also in our GO term enrichment analysis) in mayfly ovaries.

(also, methods for Fig2c clustering are not described).

AR: Methodology has been added in '4.9 Gene clustering and heatmap plots': "To generate heat maps representing gene expression of chemosensory gene families throughout multiple samples, we used previous normalised counts (see Mfuzz normalisation using DESeq2 library in 3.1 section: Transcriptomics clustering along life cycle stages (Mfuzz)) and apply the function to the datasets:

```
cal_z_score <- function(x){  
  (x - mean(x)) / sd(x)  
}
```

```
data_subset_norm <- t(apply(Chemos_matrix, 1, cal_z_score))
```

Finally, pheatmap package (v 1.0.12) was used to generate the heat maps through the 'pheatmap' function."

L284: what exactly does 'shared' mean here? Genes in the modules that have orthologues? And if 130 are shared, is that 130 Drosophila genes or 130 Cloeon genes?

AR: The 130 genes referred to *D. melanogaster*, which corresponded to 126 orthologous gene families ancestral to both *D. melanogaster* and *C. dipterum* (in Cdp this corresponded

to 128 genes). We clarified now this in the text and talked about 126 orthologous gene families: “A total of 126 orthologous gene families were shared between these modules, defining a core set of genes associated with this organ since the last common ancestor of pterygote insects (corresponding to 130 and 128 genes in *D. melanogaster* and *C. dipterum*, respectively).” Moreover, we included the GO term enrichment analysis of these 130 *D. melanogaster* genes (using all the WGCNA modules and the WingDisc module as backgrounds in topGO) in the Supplementary Table 13.

And this is out of how many genes in each of the modules?

AR: The wing module of *C. dipterum* has a total number of 699 genes (415 with *D. melanogaster* orthologs) and the wing disc module of *D. melanogaster* has 1919 genes (1418 with *C. dipterum* orthologs). To clarify this, we have added a new sheet in Supplementary Table 12 with the total number of genes per module, as well as the number of orthologous genes between the species analysed. The total number of genes per module has been also incorporated in the boxplots from Supplementary Data 1.

L305: where is this result shown? Also, where are the methods that describe exactly how tau indices were calculated?

AR: We have added now a new supplementary Table (Supplementary Table 10) in which we included tau indexes calculated from cRPKMs across RNA samples (we used only one of the two replicates per sample) and the "Main tissue" of expression for those which tau index is higher than 0.8. We are sorry for this omission in the previous manuscript version. There is now a section in the Methods explaining the methodology to obtain tau indexes.

L316: ‘clustering’ method seems not to be described anywhere (like other heatmaps), yet the fact that these two modules cluster together is being presented as a key result – methods must be detailed.

AR: We included a section in Supplementary Information describing how clustering and heatmaps were generated (See Supplementary information 4.9 Gene clustering and heatmap generation)

L319: the overlap analysis presented at the end of the supplement is very brief, please make sure this is fully described, and confirm that 42/98 is statistically significant.

AR: We included more details of the analysis in the Supplementary information that together with the scheme and example already shown in Supplementary Figure 6 will help the readers. Regarding a statistical test to confirm the significance of our gill/wing pad relations, this is not a straightforward thing to test, or at least it is not obvious to us what would be the right statistical test to use. We have taken the following approach: there were 11 tissues tested against wings and a total of 98 genes shared by wing plus another tissue. This gives an expected value of 8.9 genes per tissue. This means the enrichment ratio (obs/exp) in gills is 4.7. To assess the statistical significance, we randomized the counts of the 98 genes among the 11 tissues and scored how many times one tissue had a similar differential with the second highest sample comparable to what we observed for gills (i.e. 20 genes more than 10dpf embryos) and/or reached ≥ 42 counts in 1,000,000 iterations. We did not

observe any of these two conditions in the 1,000,000 iterations and we therefore propose a $p \leq 10^{-6}$.

L355-356: 'very specific genomic changes' This is very vague, and what exactly is the discussion point here? Probably just remove this sentence.

AR: We removed the sentence

L409: please clarify sample counts – table seems to give 37, 13 in 2x replicate, 1 with no replicate, then 10 heads.

AR: We clarified in Supplementary Table 6 sample description, especially the number of replicates for each of them and again, we apologise because of the mistake in the number of samples described, in this work we generated 37 RNA-seq samples and not 35. Moreover, we think that it is clearer now (especially in the Supplementary Table 6) that the 10 heads are replicates of three stages (4 early nymphs, 2 mid nymphs and 4 late nymphs)

Supplementary tables: the numbering has gone awry with these tables, and the text refers to Table S17 whilst there are only 15 supplementary tables. To reduce the number of supplementary tables that are currently separate files and require some back-and-forth for the reader to cross reference text and table data, I would suggest that smaller tables e.g. those that would fit on a single page, be incorporated into the main supplementary materials document.

AR: We apologise for the errors in the numbering of the tables. We have corrected them now. Regarding the inclusion of the small tables into the main supplementary materials, we have not done it since these tables are not the most informative ones, therefore, it would not improve the readability of the manuscript and it would generate two different set of tables. However, if the Reviewer has a specific suggestion on which tables could go to the main material, we would be happy to follow it.

Methods: the level of detail provided in the main text is extremely variable, e.g. opsin identification and analysis gets a rather detailed summary while other key methodological aspects are completely ignored such as GO term assignment and enrichment analyses. Please harmonise the methods summary descriptions in the main text so that nothing is ignored, and include explicit references to the further details provided in the supplement.

AR: The methods section included in the main text was just a reduced summary, since for simplicity and space constrictions it was not possible to include there the full methods, but admittedly the different aspects were not well balanced, so we have modified the methods section according to the Reviewer's suggestions. We have included some missing sections and added explicit mentions to their corresponding chapters in the supplementary information.

Reviewers' Comments:

Reviewer #4:

Remarks to the Author:

The authors have performed the requested supporting analyses to convincingly demonstrate the robustness of their clustering and hence their overall results and interpretations thereof. They have also satisfactorily updated the descriptions of their analyses to include the required relevant details. I list several mostly minor outstanding points below. One aspect, relevant to Figure 1 and Figure 4, is that updates to the figures are not accurately reflected in the text, i.e. some statements in the text that were based on previous versions of the figures seem to be no longer directly supported by the new versions of the figures, so some careful checking needs to be carried out to harmonise the main messages. Overall, I reiterate my round-1 review appraisal that this study is elegant and the results are enlightening, and now thanks to the revisions I am convinced that the results are also robust to the applied methods.

Thank you for adding the requested details to the supplement, it is now much more convincing as a clear and well-documented account of all the different stages of each analysis.

Thanks too for performing the extra analyses which convincingly show the robustness of the clustering, I was particularly pleased to see that the module preservation analyses with WGCNA were able to pull out the same signals.

With respect to the request to delete some sections from the supplement containing extra information that was not explicitly referred to in the main text, I leave this for the editor to decide. Normally the supplement contains only information that further supports or explains points made in the main text.

L134-139: The phrasing here is a bit disingenuous, "gene completeness of the genome was estimated to be 96.77% and 98.2%," in fact these values refer to complete+partial. It is a minor issue, but it would be good to see transparent reporting. Also check the numbers in the supplement "98.7% of the insecta_odb9 BUSCOs are present" but if 1.8% are missing then surely present=98.2%?

Thanks for running the pseudohaploid check, this is reassuring and provides good support for an assembly free of haplotypes.

L140: seventh => seven

L144: Table S5 – check formatting of sheets, the first one is OK but others data are strings not separated into cells

SOM page 8: "The distribution was plotted (Supplementary Figure 1B) using the helper scripts from RepeatMasker" should this point to 1D rather than 1B?

L152: I'm happy with the replicate info in the legend – thanks. Having the Mfuzz clusters in colour makes it much clearer – thanks. You might want to reference Supplementary Table 6 here otherwise it is only referenced from legend.

Thanks too for adding the gene numbers to Fig1g and FigS2, it really helps.

Figure 1 and lines 165-176: It is good to see many of the same terms appearing after the switch to using topGO. However, there are some changes and these have not been conveyed in the corresponding text. E.g. it is strange to specifically mention in the text terms like 'ectoderm development' when they do not appear in the figure (this term did appear previously, so I can see why it is in the text, but now the text does not accurately reflect the new results). Please thoroughly check the statements made in the text to make sure they make sense with respect to the new topGO results, otherwise it is incredibly confusing for the reader. E.g. the text suggests Cluster 9 should have some chitin/cuticle terms – I cannot see any in this cluster. Also, I couldn't help but notice an indication of an immune response in Cluster 18 (not there previously), is this worth mentioning?

SOM page 10: "We also determined that the overlap between core genes in each of the cluster was zero." => "We also determined that the overlap between core genes in each of the cluster was zero."

Table S8: one sheet is named 'Hoja30'

Figure 2: panel b still refers to NMDAs and CD36s/SNMPs, which, as far as I can tell, are otherwise not mentioned anywhere in the main text, please either briefly explain what they are and why they are there or remove them.

Figure 2: it is useful to now have the gene numbers in the figure (panel b), but please state in the legend what the numbers in parentheses are.

L306: if it is called 'wing pad' in the text then it should be also labelled the same in the figure? As far as I can see in the figure it is labelled simply as 'Wing'.

Figure 4a: This is different from the previous version. Most of the significant pairs seem to be the same (but for example the weak Cdip Muscle – Dmel Testis association has now gone), but the dendrograms are much better separated and re-ordering makes it visually easier to understand. What I don't understand is what changed in the analyses that brought this about? From reading the responses I thought that while replicates with different parameters were performed to confirm robustness of the modules, the main analysis itself for defining modules did not change? Is it just a question of visualisation (but then why are the dendrograms different)? This is also true of the equivalent plots in Figure S6.

L311-312: The previous version of Fig4c did show some terms that were more obviously related to wing development (e.g. 'wing disc development') – so what has changed here? Similar to the point above, I understood that modules were redefined with different parameters just to establish robustness, but this makes it seem like the actual main results have changed somewhat – I could not find what might have changed. In any case, as for Figure 1, please make sure that statements made in the text accurately reflect what is being shown in the figures since the updates – otherwise readers will be left confused and possibly in doubt over the findings.

SOM page 20: "through the 'pheatmap' funcion" => "through the 'pheatmap' function"

L358: "with respect of" => "with respect to"

L361: "adults relying" => "adults rely"

L407: "associated to developing wing" => "associated with wing development"

Thank you for balancing out the methods descriptions, this section is much better now.

Response to Reviewer R2

Reviewer #4 (Remarks to the Author):

The authors have performed the requested supporting analyses to convincingly demonstrate the robustness of their clustering and hence their overall results and interpretations thereof. They have also satisfactorily updated the descriptions of their analyses to include the required relevant details. I list several mostly minor outstanding points below. One aspect, relevant to Figure 1 and Figure 4, is that updates to the figures are not accurately reflected in the text, i.e. some statements in the text that were based on previous versions of the figures seem to be no longer directly supported by the new versions of the figures, so some careful checking needs to be carried out to harmonise the main messages.

Overall, I reiterate my round-1 review appraisal that this study is elegant and the results are enlightening, and now thanks to the revisions I am convinced that the results are also robust to the applied methods.

AR: We want to thank again the Reviewer for appreciating the impact of our work and his/her thorough and constructive revision.

Thank you for adding the requested details to the supplement, it is now much more convincing as a clear and well-documented account of all the different stages of each analysis.

AR: We would like to thank the Reviewer for this comment, we agree that the methodology used is much clearer now.

Thanks too for performing the extra analyses which convincingly show the robustness of the clustering, I was particularly pleased to see that the module preservation analyses with WGCNA were able to pull out the same signals.

AR: We are glad that the Reviewer appreciates our efforts to show the robustness of our analyses

With respect to the request to delete some sections from the supplement containing extra information that was not explicitly referred to in the main text, I leave this for the editor to decide. Normally the supplement contains only information that further supports or explains points made in the main text.

AR: As we pointed out in our previous response, we believe these sections contain valuable information that could be of interest for some readers. However, we will also follow any recommendation from the editors on this regard.

L134-139: The phrasing here is a bit disingenuous, “gene completeness of the genome was estimated to be 96.77% and 98.2%,” in fact these values refer to complete+partial. It is a minor issue, but it would be good to see transparent reporting. Also check the numbers in the supplement “98.7% of the insecta_odb9 BUSCOs are present” but if 1.8% are missing then surely present=98.2%?

AR: We modified the sentence to clarify these numbers: “...and 98.2% (94.1% Complete Single-copy, 2.8% Complete Duplicated and 1,3% Fragmented BUSCOs), according to...” and we corrected the mistake in the Supplementary information, the correct number is 98.2% as stated in the main text and the Supplementary Table 1.

Thanks for running the pseudohaploid check, this is reassuring and provides good support for an assembly free of haplotypes.

L140: seventh => seven

AR: It has been changed

L144: Table S5 – check formatting of sheets, the first one is OK but others data are strings not separated into cells

AR: We have modified the format

SOM page 8: “The distribution was plotted (Supplementary Figure 1B) using the helper scripts from RepeatMasker” should this point to 1D rather than 1B?

AR: The Reviewer is right, it has been corrected.

L152: I’m happy with the replicate info in the legend – thanks. Having the Mfuzz clusters in colour makes it much clearer – thanks. You might want to reference Supplementary Table 6 here otherwise it is only referenced from legend.

AR: We have included now a reference to the Supplementary Table 6.

Thanks too for adding the gene numbers to Fig1g and FigS2, it really helps.

AR: We thank the Reviewer for this suggestion

Figure 1 and lines 165-176: It is good to see many of the same terms appearing after the switch to using topGO. However, there are some changes and these have not been conveyed in the corresponding text. E.g. it is strange to specifically mention in the text terms like ‘ectoderm development’ when they do not appear in the figure (this term did

appear previously, so I can see why it is in the text, but now the text does not accurately reflect the new results). Please thoroughly check the statements made in the text to make sure they make sense with respect to the new topGO results, otherwise it is incredibly confusing for the reader. E.g. the text suggests Cluster 9 should have some chitin/cuticle terms – I cannot see any in this cluster. Also, I couldn't help but notice an indication of an immune response in Cluster 18 (not there previously), is this worth mentioning?

AR: We have changed the text to accommodate it to the new topGO term enrichment analysis, we are sorry for missing this. We refer now to terms that are present in Figure 2 (previous Figure 1g) Regarding, cluster 9, although now it does not show specifically chitin/cuticle terms (thanks for spotting this), it exhibits enrichment in high metabolism and respiratory process, that are also consistent with the continuous moulting process occurring at these periods. However, these terms are probably less worth mentioning than the chitin, so we have now removed the reference to cluster 9 from the main text. A relationship between moulting and immune response has been previously described in the literature, thus, it is also consistent to find also these terms enriched in our analyses, as the Reviewer noticed. We have included "...such as chitin-based cuticle development and defence response^{19,20}."

SOM page 10: "We also determined that the overlap between core genes in each of the cluster was cero." => "We also determined that the overlap between core genes in each of the cluster was zero."

AR: Changed

Table S8: one sheet is named 'Hoja30'

AR: Corrected

Figure 2: panel b still refers to NMDAs and CD36s/SNMPs, which, as far as I can tell, are otherwise not mentioned anywhere in the main text, please either briefly explain what they are and why they are there or remove them.

AR: We have mentioned now in the main text these gene families.

Figure 2: it is useful to now have the gene numbers in the figure (panel b), but please state in the legend what the numbers in parentheses are.

AR: It has been added to the figure legend

L306: if it is called 'wing pad' in the text then it should be also labelled the same in the figure? As far as I can see in the figure it is labelled simply as 'Wing'.

AR: We have replaced 'Wing' by 'wing pad' in the figure.

Figure 4a: This is different from the previous version. Most of the significant pairs seem to be the same (but for example the weak Cdip Muscle – Dmel Testis association has now gone), but the dendrograms are much better separated and re-ordering makes it visually easier to understand. What I don't understand is what changed in the analyses that brought this about? From reading the responses I thought that while replicates with different parameters were performed to confirm robustness of the modules, the main analysis itself for defining modules did not change? Is it just a question of visualisation (but then why are the dendrograms different?)? This is also true of the equivalent plots in Figure S6.

AR: Yes, this is just a question of visualization, the main analysis itself for defining modules has not changed at all, we had only improved the heatmap visualization and how we performed the clustering associated to this heatmap. We had detailed all this in the corresponding WGCNA section, but we should have also explained it in the previous Response to the Referees, we apologize for this. Prompted by the previous Reviewer comments on some of the weak associations such as the testis one, we re-plotted again the heatmaps trying to get a better visualization of the results, given that a few extreme p-values (the two top most significant module comparisons reach a $-\log(\text{pvalue})$ of 100 when most of the other significant comparisons are under 10) dominate the colour range of the heatmap, making the colours in the figure more difficult to visualize. Thus, as explained in the Supplementary Information, we clipped the values at 10, effectively flattening the colours of those module comparisons above that value. Similarly, we set a minimum value of 3 for the visualization of the heatmap, to only show module associations that were highly significant ($\text{pval} > 0.001$); since we think it was important to be stringent even if this meant that some weak (and likely biologically meaningful) associations such as the Cdip Muscle – Dmel Testis are not visible now. While doing this, we realised that these extreme values were also dominating the clustering and that if we performed the clustering after clipping the values at 10 rather than clustering and clipping afterwards, the dendrograms were much better separated and the clustering was much more biologically meaningful and easier to understand, as the Reviewer acknowledges.

L311-312: The previous version of Fig4c did show some terms that were more obviously related to wing development (e.g. 'wing disc development') – so what has changed here? Similar to the point above, I understood that modules were redefined with different parameters just to establish robustness, but this makes it seem like the actual main results have changed somewhat – I could not find what might have changed. In any case, as for Figure 1, please make sure that statements made in the text accurately reflect what is being shown in the figures since the updates – otherwise readers will be left confused and possibly in doubt over the findings.

AR: This are the same type of GO term changes that the Reviewer has also commented above for Figure 1 and lines 165-176: we have to re-run this analysis to keep consistency with the rest of the experiments (topGO instead of DAVID), following

the Reviewer's suggestion for GO analyses. This is why there have been some slight changes in the enrichment terms, however, these changes are really minor and the terms that we obtained are still related to wing development, as 'establishment of tissue polarity', 'morphogenesis of a polarized epithelium', 'open tracheal system development', 'regulation of tube size', etc. are all terms/processes involved in insect wing development. We have checked the main text to make sure that it is consistent with what we show in the figure.

SOM page 20: "through the 'pheatmap' funcion" => "through the 'pheatmap' function"

AR: Changed

L358: "with respect of" => "with respect to"

AR: Done

L361: "adults relying" => "adults rely"

AR: Changed

L407: "associated to developing wing" => "associated with wing development"

AR: Changed

Thank you for balancing out the methods descriptions, this section is much better now.